# New Model of Coastal Evolution in the Ria de Vigo (NW Spain) from MIS2 to Present Day Based on the Aeolian Sedimentary Record

Carlos Arce-Chamorro *[ID], Juan Ramón Vidal-Romaní [ID] and Jorge Sanjurjo-Sánchez [ID]

Grupo Interdisciplinar de Patrimonio Cultural y Geológico 'CULXEO', Instituto Universitario de Xeoloxía "Isidro Parga Pondal", ESCI, Campus de Elviña, Universidade da Coruña, 15071 A Coruña, Spain
* Correspondence: carlos.arce@udc.es

**Abstract:** Galician Rias are fluvial valleys that were flooded during the last marine transgression in the Atlantic margin. The study of fossil dunes in the Cies Islands, a small archipelago in the mouth of the one of the rias (Ria de Vigo), allowed us to reconstruct the coastal evolution from the end of the Late Pleistocene to the present day. During this period, sea-level was 100 metres below the present one and the shoreline located about 5–10 kilometres away. About 15,000 years ago, sea-level rise began, radically modifying the coastline. This started with a gradual advance of large dune fields on both sides of the valley. The aeolian accretion continued until the Late Holocene, finishing when the sea reached its present level.

**Keywords:** aeolian accretion; coastal dunes; OSL-dating; MIS2; Holocene; sea-level; submerged forest; Ria de Vigo; NW Iberian Atlantic coast

## 1. Introduction

The main landforms along the Atlantic coast of Galicia (inset in Figure 1) are represented by tectonic cliffs and rias affected by neotectonic uplift [1]. As to Galician rias (the western estuaries are indicated in the inset of Figure 1), these peculiar formations have a specifically continental origin [2] as fluvial valleys flooded during the Holocene transgression. With the exception of fluvial sediments, the sedimentary record during the Quaternary on this rocky coastal border is essentially represented by pebble, gravel and coarse sand deposits on rocky platforms at different elevations [3,4], which correspond to old marine terraces. These coarse-grained sediments are also represented on the present-day coastline by shingle beaches, although their formation and analysis are not discussed in this paper. The other type of sediments relates to aeolian sands that correspond to small but numerous outcrops distributed along the NW Iberian coast [5] (see some examples in Figure 2 and the aeolianite outcrop studied here (Figure 3)). Initially, these aeolian sands were considered erroneously as beach deposits [6] and were used to define Quaternary sea-levels along the Galician coast. However, an aeolian origin is now assumed [5] for this type of sediment.

Until now, the knowledge of sediments both inside the rias and on the continental shelf were based on high-resolution seismic-reflection profiles, complemented with deep-core or vibrocorer drillings [7–12], in which micropalaeontological studies were not performed. On the other hand, the chronology of the proposed sedimentary units was based on scarce radiocarbon dating of organic remains assuming only a few suspected Pleistocene fluvial sediments. Regarding the Upper Pleistocene fluvial deposits in the Ria de Vigo, these authors describe upper sedimentary units at the limit of the radiocarbon dating by accelerator mass spectrometry (40–45 ky), overlying suspected Tertiary deposits [13,14], with no other absolute ages beyond [14]C limit. In this sense, these infills in the Ria de Vigo would be positively related to the nearest fluvial deposits in the basin of the lower Miño

River (Pontevedra) (inset in Figure 1) dated by optically stimulated luminescence (OSL) and ascribed to the Upper Quaternary [1], as similar sediments of continental origin dated in the coast of Galicia and northern Portugal [15–19], although none of these data have been considered in previous assumptions.

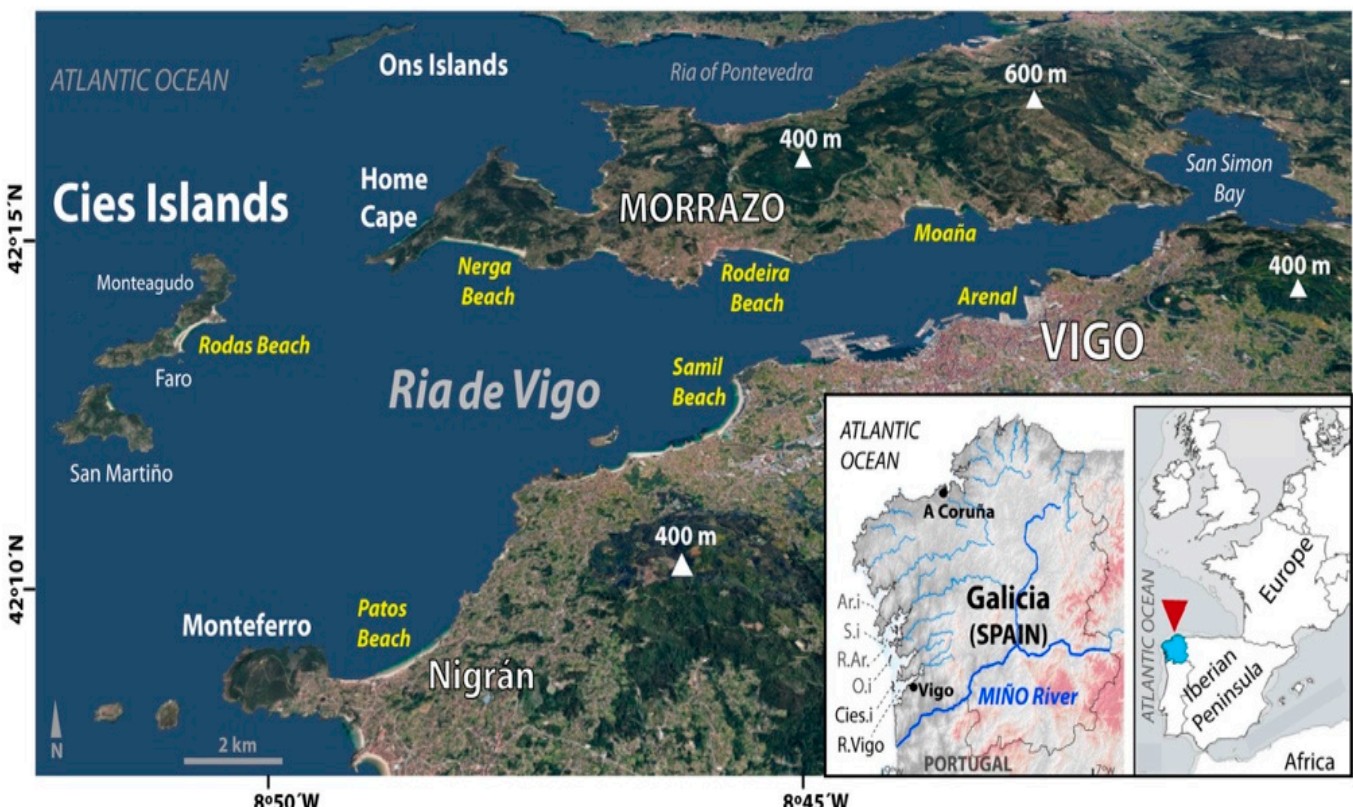

**Figure 1.** Present day configuration of the Cies Islands and Ria de Vigo showing the main sand beaches (Rodas, Patos, Nerga, Samil, Rodeira and Moaña) related to dune fields and climbing dunes, as examples of aeolian deposits. R.Vigo: Ria de Vigo; Cies.i: Cies Islands; O.i: Ons Islands; R.Ar.: Ria de Arousa; S.i: Sálvora Island; Ar.i: Areoso islet.

Regarding the sand deposits, the remaining sedimentary cover is mostly represented by sands of less than 15 ky, which have been interpreted only as deposits of marine origin, although neither saltwater diatoms nor foraminiferal associations are observed. Conversely, the data presented here suggest that they are of aeolian origin, similar to the current transgressive aeolian sand-sheets [20] described on other coasts of the world today, but mobilised by wind since the end of the Last Glacial period from a lower sea-level than in the present. Under this new hypothesis, the different levels reached by the sea in this Atlantic coast during this time period have not been satisfactorily understood, thus misunderstanding when the Galician rias were emerged or when the ocean waters flooded them again.

These aeolianite outcrops preserved in the coast of Galicia (Figure 2) are well-sorted azoic deposits that are mainly represented by a thin siliciclastic wedge less than 5 m thick [5], and they have been interpreted as climbing dunes correlated to a "regressive marine stage" [5]. Sedimentary structures, such as cross-stratification, have not been clearly preserved in these aeolian formations (Figure 3), something frequent, but not generalised, to all climbing dunes [5]. This is also the case of the aeolianite outcrop of Cies Island studied here (Figure 3c). However, these sedimentological issues are not discussed in the present paper.

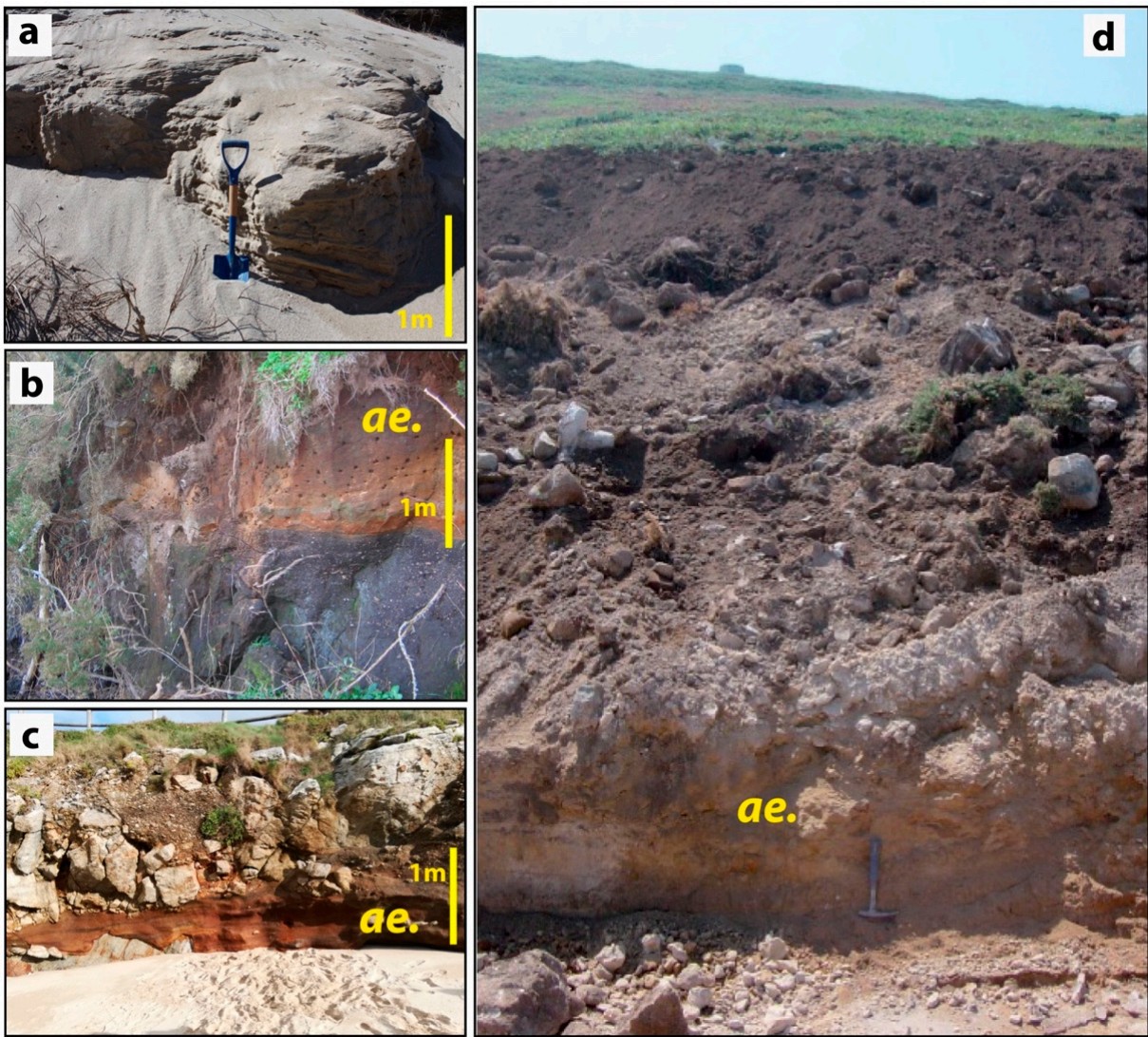

**Figure 2.** Some examples of fossil dunes preserved along the Galician coast and interpreted as climbing dunes [6]. (**a**) Climbing dune from Trece (43°11′15″ N, 9°08′45″ W; WGS84); (**b**) 30 ky old aeolianite from Tal (42°47′16″ N; 9°00′55″ W) [16]; (**c**) Baldaio aeolianite (43°18′11″ N; 8°39′24″ W); (**d**) Penaboa aeolianite (43°22′50″ N; 8°26′18″ W), dating back ≈ 300 ky [4]; ae: aeolian sands.

The age of these sands (Figure 3), estimated by optically stimulated luminescence (OSL) [15,16] or infrared stimulated luminescence (IRSL) [21], can be established for all equivalent deposits located on the entire coast between 300 ky (aeolianite of Penaboa, A Coruña, Galicia, Spain) [4] and 12 ky [22], which are roughly related to regressive marine stages. Regarding the fossil dunes formed on the Galician coast during marine isotope stage 2 (MIS2), it is important to highlight that the sea-level rise of about 120 m during the last 15,000 years in the study area meant the disconnection due to marine flooding from their source area (of sand). This is confirmed by the presence of aeolian sands of 20 ky more than 100 m deep as described [12] in the continental platform near the Cies Islands (Figure 4; Table 1). Currently, under the wet and temperate climate of the North Atlantic peninsular coast [23], the most recent aeolian formations (<5 ky) are being affected by wind and sea waves even when stabilised by land vegetation [5,24,25], due to the breakage of the vegetal cover during the storms. In other cases, however, the preservation of the climbing dunes is due to the active slope dynamics developed in some steep cliff sections of the Galician coast, fossilizing the aeolianites by slope deposits [3,5] (Figure 3c,d).

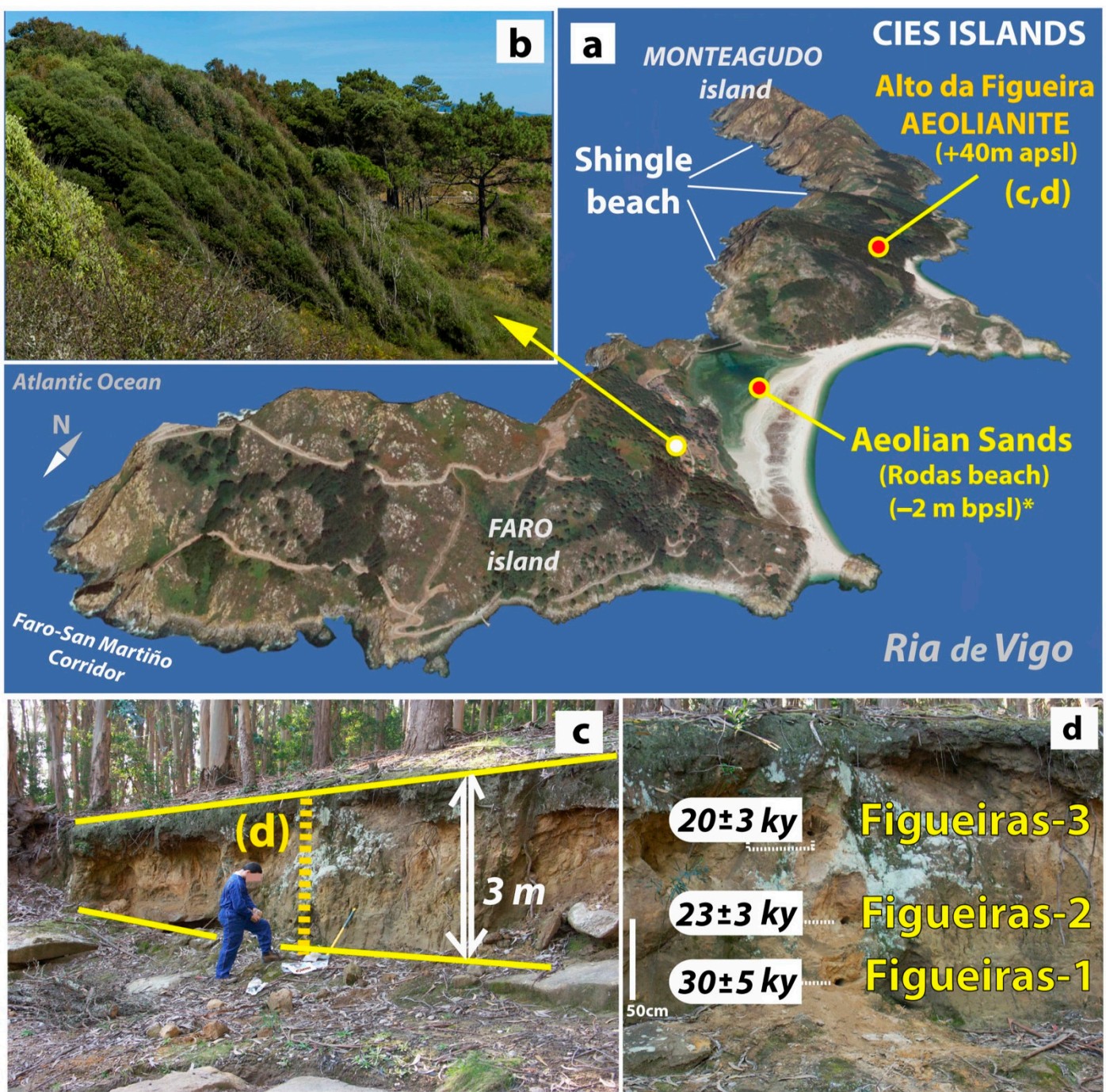

**Figure 3.** Monteagudo and Faro Islands (from the Cies Islands). (**a**) Aeolianite outcrop from Monteagudo Island (Alto da Figueira; 42°13′45″ N, 8°54′15″ W; WGS84) and aeolian sands (* [8]) location, as mainly aeolian sediments developed on the eastern slope at the end of the Upper Pleistocene. Shingle beaches are only observed in the western side. (**b**) Wind effect on vegetation on the eastern side of Faro Island, showing W–SW wind direction. (**c,d**) Fossilised climbing dune at +40 m above present sea-level (modified from [15]).

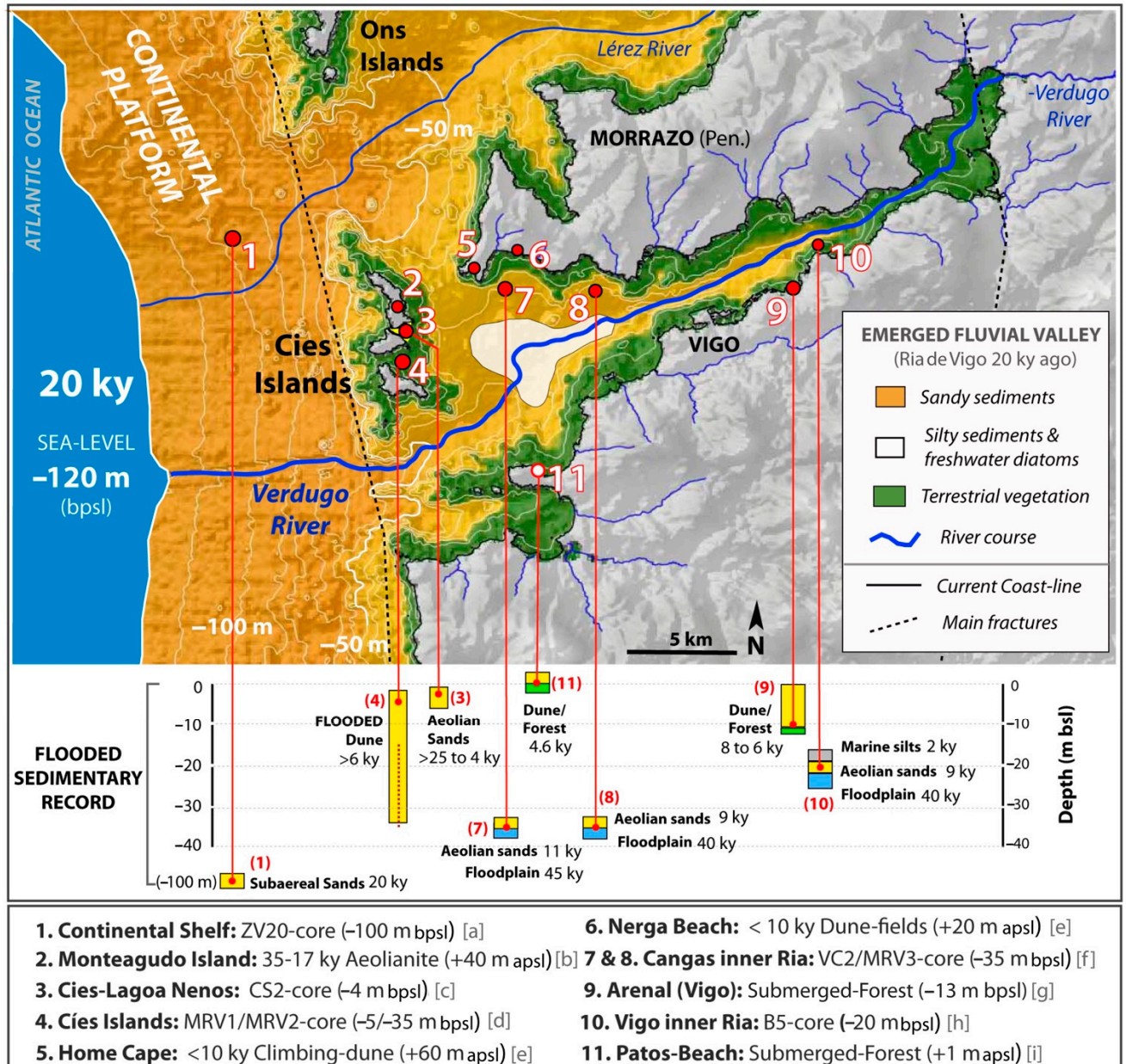

**Figure 4.** Geomorphology of the Ria de Vigo with sea-level −120 m below present sea-level (bpsl) at the end of the Last Glacial episode, showing the hypothetical position of the Verdugo River through the emerged fluvial valley; bathymetry from [26]. Green strip shows the advance of terrestrial vegetation and yellow area shows the predominance of sands at the bottom surface [7]. The white area represents silty sediments with freshwater diatoms [27]. Red dots show the main sand deposits related to aeolian sediments—some of them described as marine sediments (7–10) and reinterpreted in this paper as aeolian sands based absolute chronology and depth—[a]: [12]; [b]: [15]; [c]: [8]; [d]: [28]; [e]: [5]; [f]: [10]; [g]: [29]; [h]: [10]; [i]: [30]. Digital elevation database from [31]; apsl: above present sea-level.

**Table 1.** Conventional [14]C ages from sedimentary deposits related to flooded aeolian sands in the Cies Islands and Ria de Vigo. All these data have been recalibrated in this paper using the INTCAL20 curve [32] from the on-line OxCall program (v4.4), except for the calibrated age of the Patos forest (*) from [30]. The table specifies the dated material and the type of sediment in which it was found.

| Sequence Location | Depth (m) | 14C Uncal. Age (BP) | INTCAL20 cal yr (BP) | Material | Reference |
|---|---|---|---|---|---|
| Platform shore | 100 | 18,010 ± 90 | 22,186–21,466 | Mixed forams (subaerial sands) | [12] |
| Barrier lagoon (Rodas beach/Cies) | 2 | 21,680 ± 60 | 26,030–25,835 | Organic matter (sand dunes) | [8] |
| | 2 | 17,240 ± 50 | 20,933–20,604 | Organic matter (sand dunes) | |
| | 2 | 3750 ± 30 | 4233–3916 | Organic matter (sand dunes) | |
| Flooded dune (Cies) | 5 | 5850 ± 30 | 6744–6561 | Shell fragment (bioclast in sands) | [28] |
| | 35 | 3220 ± 30 | 3482–3376 | Shell fragment (bioclast in sands) | |
| Inner ria | 35 | 39,890 ± 440 | 44,086–42,628 | Bryophyte (floodplain) | [9,10] |
| | 35 | 9960 ± 40 | 11,682–11,250 | Organic matter (sands) | |
| | 20 | 8010 ± 40 | 9010–8655 | Shell fragment (bioclast in sands) | |
| | 20 | 1490 ± 30 | 1055–1001 | Shell fragment (bioclast in silts) | |
| Patos forest (mouth of the ria) | 0 | (*) | 4600 * | Organic matter (fossil forest soil) | [30] |
| Vigo–Arenal forest (bottom of the ria) | 13 | 7070 ± 30 | 7967–7836 | Wood (whole trunks and roots) | [29] |

In previous studies, the referred aeolian deposits on the coast of Galicia were mistakenly identified as marine sands although recently reworked by wind [33,34]. Conversely, sedimentological analysis of old climbing dunes [5] and chronological data [16] suggest that these sand deposits were formed when sea-level was much lower than today, being transported inland by the wind. Subsequently, as sea-level rose, the dunes were flooded or breached by waves, as it happens with sediments reworked by the sea. This is also suggested by both the grain size and impact marks on quartz grains observed for such deposits from a nearby ria at depths of 45 to 75 m [35]. However, some papers assumed such sand as old marine interglacial levels [36,37], while they were clearly identified as aeolian sands in some others [38,39]. These latter authors interpreted such deposits as remains of aeolian formations alternately deposited and eroded during the glacioeustatic oscillations of the Pleistocene. A complete sedimentological study of six outcrops of coastal aeolianites in Galicia carried out by [5] included, for the first time, grain size, morphoscopic, mineralogical and textural analysis by SEM (Scanning Electron Microscopy). All of these homogeneous sandy outcrops are constituted by fine–medium (98%) and subrounded (<95%) quartz grains (>90%), showing both V-shaped, arc-shaped and conchoidal or Hertzian fractures on grains, which support a true aeolian origin [5].

Regarding to the coastal dunes in the Cies Islands, previous studies show that they were formed from 25 ky to 4 ky [8] (Table 1). Such dune fields are located on the present coastline linked to sand beaches (as Rodas, Nerga, Patos and Samil beaches) (Figures 1, 3 and 4). The main objective of this work is the review of the aeolian sands series (Figure 4) based on their chronological, bathymetric and micropalaeontological data in the Ria de Vigo (Table 1). This would allow us to reinterpret the evolution of the ria from the end of MIS2 to the present day, which could be extrapolated to the Galician Rias based of the latest model of coastal evolution during the Holocene [4].

## 2. Study Area

The Ria de Vigo (Figures 1 and 4), as the Galician Rias, is a recent landform defined as a primary coast type [2:98]. During the present interglacial, marine waters flooded the lowermost areas of the Verdugo–Oitavén rivers drainage basin (of 710 km$^2$). The Ria de Vigo is the most southern ria of the coast because the Miño River, further south, is not a ria but an estuary [1] (see inset in Figure 1). This ria develops longitudinally in a WNW–ESE direction for 30 km, with the Cies Islands located at its mouth (Figure 1). Previous authors [7,40] identified shallow sedimentation (probably Holocene) overlying Pleistocene units of greater thickness. In the inner section of the ria there is a greater presence of silts and silty sands, while in the central and outer sections there is a predominance of sand-sized sediments [7,12,40] (Figure 4).

Regarding to the nearest shore sedimentation sequences (Figure 4), previous authors [12,41] reported (i) silt and very fine sand hypothetically carried by the marine currents from south to north [41] and (ii) fine, medium and coarse sands from the break-up of coastal sandy beaches during the storm events [41]. They also describe the occasional presence of gravels hypothetically transported by the main drainage networks (e.g., rivers Douro (Porto, Portugal), Miño-Sil (Galicia–Portugal) (Figure 1), Verdugo–Oitavén in the Ria de Vigo or Lérez in the Ria de Pontevedra (Figure 4), among others). These coarse-size materials were most likely reworked during the glacioeustatic Quaternary cycles [4]. For about 40 km, this platform has a slope of about 1% down to a depth of 200 m. Thereafter, it abruptly drops to depths of 1000 m through the continental slope.

The Cies Islands are located between the Ria de Vigo and the continental platform, belonging a tectonic alignment (Figure 4) of Cenozoic age [42]. This archipelago of Cies are constituted by three islands (Monteagudo, Faro and San Martiño) that show an N–S alignment about 4.5 km$^2$, reaching an altitude of up to 197 m above present sea-level (apsl). These islands are located in the northern half of the mouth of the Ria de Vigo. The northern foothills of the archipelago are 2.5 km away from the cape Cabo Home (Peninsula de Morrazo). These bottoms (Figure 4), up to 25 m deep, are covered by sands [7] describing a flat relief [26]. The southern foothills are 5 km from the coastline (Monteferro) (Figures 1 and 4), describing an irregular submarine relief up to 50 m depth [7,26] between granitic outcrops (Figure 4). At the base, a sand-dominated sedimentary cover of up to 20 m was described [7,12]. The topography of the Cies Islands offers a clear asymmetry, with a western face of steep slope due Cenozoic faulting (see main fractures in Figure 4). On the western side, shingle beaches are now exposed below the sand cover (Figure 3). On the east side of the Cies Islands, more sprawling and protected, the massive outcrop of aeolianites of Alto da Figueira (Monteagudo Island) (Figures 1, 3 and 4) is preserved at +40 m (apsl). The extent of this aeolian formation cannot be accurately detailed [15] and the only evidence available is the 3 m aeolianite outcrop shown in Figure 3.

## 3. Materials and Methods

For this study, three samples were taken of the aeolianite outcrop from Alto da Figueira at Montefaro Island (Cies Islands) (42°13′45″ N, 8°54′15″ W; WGS84) (Figure 3). Grain size was assessed by dry sieving or the raw sample (10 g). Morphological analysis was assessed with stereoscopic microscopy (50 quartz grains per sample). This information is complemented with an analysis of the surface of the most representative quartz grains of

the samples and in better preservation conditions by Scanning Electron Microscopy (SEM) (JEOL mod. JAM-6400) by secondary electrons detection (25 Kv). For this, vacuum Au metallisation (0.05 mBa) of the cleaned and dehydrated samples is carried out by cathodic electrospray (Sputter Coater) (BAL-TEC SCD004).

OSL sampling was done by hammering three cylindrical steel-cores extracted along the sediment profile at depths of 200, 140 and 80 cm (distance from the top to the bottom of the outcrop). The outer part of the cores was removed under subdued red light in the Luminescence Laboratory of the University of A Coruña, and the central part dried and sieved. Coarse sand grains (180–250 µm) were treated with HCl and $H_2O_2$. Feldspars and heavy minerals were removed by density separation with sodium polytungstate solutions (densities of 2.62 and 2.70 g/cm$^3$) and the obtained quartz was etched in concentrated hydrofluoric acid (40%) to remove any remaining feldspars. This etching step removed approximately 10% of the beta dose rate ($D_r$) [43]. The quartz grains were checked with infrared (IR) stimulation to ensure the absence of minerals other than quartz. Luminescence measurements were performed on multigrain aliquots (2 mm in diameter) mounted on stainless steel discs in a Riso-DA15 automated TL/OSL reader equipped with blue light emitting diodes (LEDs) (470 ± 30 nm) for stimulation and a 9235QA photomultiplier. A Hoya U-340 filter was placed between the photomultiplier and the samples. To irradiate the samples, beta doses were used, using a $^{90}Sr/^{90}Y$ source that provided a dose rate of 0.120 ± 0.003 Gy/s. To estimate the equivalent dose ($D_e$), the Single-Aliquot Regenerative dose (SAR) protocol was used after performing preheat tests, and recovery tests were carried out [44]. The OSL signal was stimulated during 40 s, with the last 4 s of decay-curve used for subtraction of the fast component (first 0.4 s of stimulation) to determine the OSL-signal [45], known as Late Background (LBG). The Early Background (EBG) was also used to subtract the background signal from 0.5 to 4 s of decay-curve [46], with both signals compared.

The $D_r$ was estimated using low-background gamma spectrometry on bulk samples. Marinelli beakers were used and measurements were taken in a coaxial Camberra-XTRA gamma detector (Ge-Intrinsic) model GR6022 within a 10 cm thick lead shield. A gamma cocktail solution (manufactured by LMRI, CIEMAT, Madrid, Spain) was used for calibration, with the spectrometer yearly intercalibrated with CSN (National Nuclear Security Council of Spain) and IAEA. Gamma spectrometry reports have assumed the activities of some isotopes to be those of their parents considering the equilibrium groups [47,48]. Guérin's conversion factors were used [49]. The alpha contribution was neglected for quartz dose rates, the beta dose-rate being corrected due to the HF etching step. An internal dose rate contribution from U and Th in the quartz grains of 0.02 ± 0.01 Gy/ka was assumed [50]. Water content and water saturation values were assessed in the laboratory for all samples to estimate average water content, and the cosmic dose rates were calculated in accordance with Prescott and Hutton [51].

## 4. Results

### 4.1. Grain Size, Morphology and Microscopic Analyses

The grain size shows greater than 95% medium and fine sand [52,53] in all three samples (Figure 5a). The symmetrical trend of the grain size distribution and good sorting fits well with most aeolian sands [5]. Morphological analysis mostly shows subrounded grains, following the criteria of [54]. Microscopic analysis by SEM clearly shows aeolian impact marks (Figure 5b).

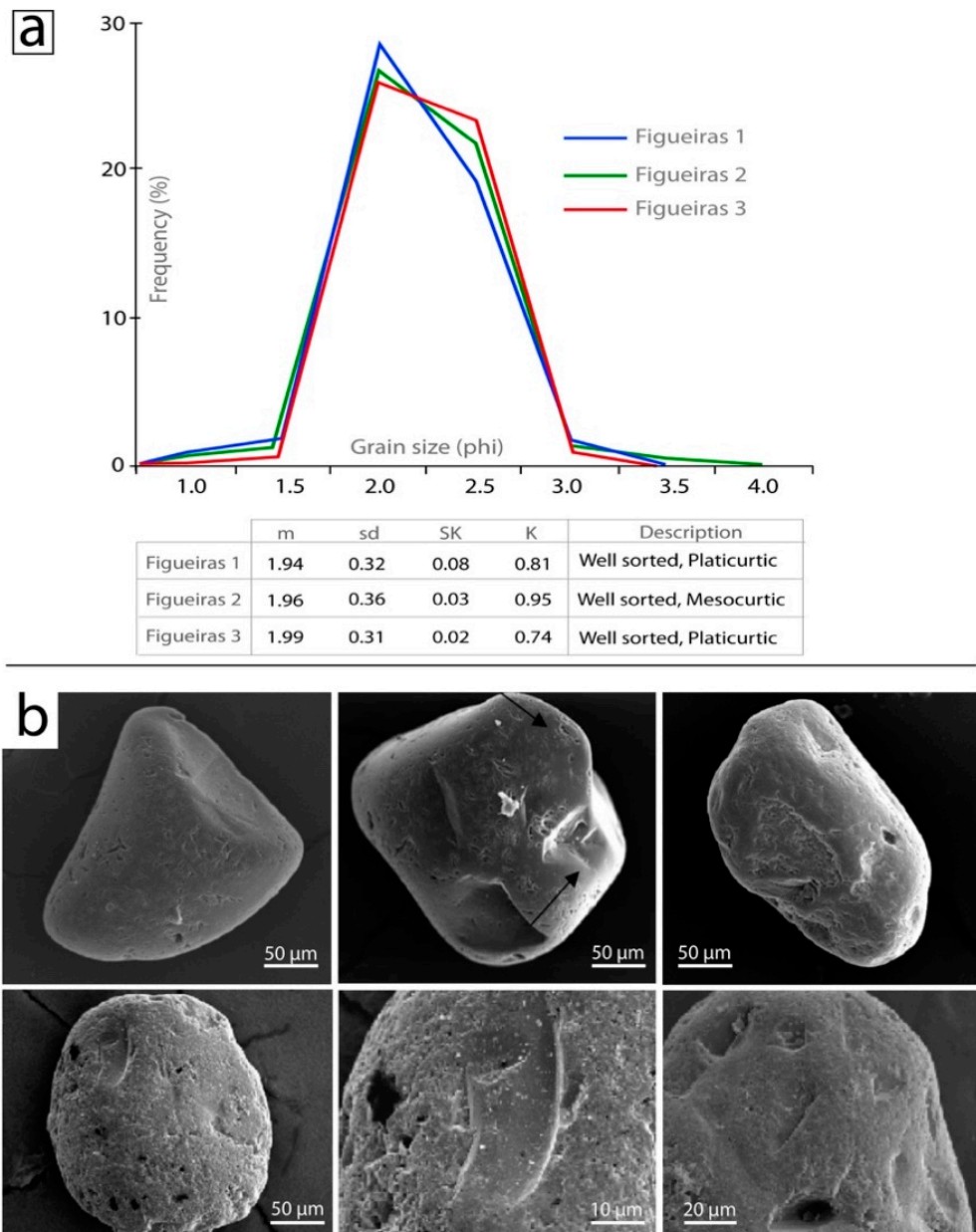

**Figure 5.** (**a**) Dry-sieving particle size and distribution of the samples from Figueiras aeolianite (Cies Islands): (m) mean, (sd) standard deviation, (SK) skewness coefficient, (K) kurtosis from the method described in [55]. (**b**) Scanning Electron Microscope (SEM) images of quartz grains showing aeolian impact marks.

### 4.2. OSL-Dating: Sampling, Equivalent Dose ($D_e$) and Dose Rate ($D_r$)

To estimate the dose rate ($D_r$) (Table 2), a saturation percentage of 20 ± 2% was assumed for all the samples. In this material, there is an average proportion of fine sand over 90%, so the drainage conditions are favourable, reducing the attenuation of the radiation for content in interstitial water [56]. The high grade of homogeneity minimises the variations related to beta-dosimetry [57]. For these samples, no disequilibrium is observed in the uranium and thorium decay chains, as the values of $^{226}$Ra and $^{228}$Ra are the same or very similar to those of $^{238}$U and $^{232}$Th (Table 2). Total $D_r$ of the Figueiras samples is very similar, ranging between 2.7 and 2.1 Gy/ka (Table 2) [15]. These rates are also similar to those estimated for other aeolianite outcrops [16] and quartz-rich fluvial deposits in the region [1,16–18].

**Table 2.** Dose-rate estimation of the samples from aeolianite outcrops of the Cies Islands. Radioisotopic activity of the series of $^{238}$U and $^{232}$Th ($^{226}$Ra and $^{228}$Ra values are also shown), and $^{40}$K (Becquerel/kg); Dose rate (Total-$D_r$) (Grays/ka). Sampling distance (cm) from the top to the base of the outcrop.

| Samples | $^{238}$U (Bq/kg) | $^{226}$Ra (Bq/kg) | $^{232}$Th (Bq/kg) | $^{228}$Ra (Bq/kg) | $^{40}$K (Bq/kg) | TOTAL-$D_r$ (Gy/ka) |
|---|---|---|---|---|---|---|
| Figueiras-1 (200 cm depth) | $18 \pm 2.4$ | $18 \pm 2.4$ | $20 \pm 2.6$ | $20 \pm 2.6$ | $660 \pm 38$ | $2.7 \pm 0.4$ |
| Figueiras-2 (140 cm depth) | $27 \pm 1.3$ | $19 \pm 2.1$ | $19 \pm 2.8$ | $19 \pm 2.8$ | $570 \pm 35$ | $2.4 \pm 0.3$ |
| Figueiras-3 (80 cm depth) | $19 \pm 1.2$ | $16 \pm 2.2$ | $15 \pm 4.8$ | $15 \pm 4.8$ | $485 \pm 31$ | $2.1 \pm 0.3$ |

The measured aliquots show bright OSL-signals and suitable OSL-growth curves to interpolate [58] (Figure 6), and the $D_e$s show symmetrical distributions that fit a normal distribution, although scattered (Figure 6b–d), so the Central Age Model (CAM) [59] was used to estimate the $D_e$s, summarised in Table 3. For these aliquots, $D_e$ uncertainty is lower than 10%. The Overdispersion (*OD*)—with respect to the CAM average for these samples—falls within an acceptable limit (20–30%) for samples with complete bleaching [60], with the exception of Alto da Figueira2 that has an *OD* of 50% (Table 3). Nevertheless, despite this high overdispersion, the LBG-CAM burial-age estimation is stratigraphically consistent. If we compare the CAM $D_e$s estimates using the LBG with these obtained using the EBG [46], for sample Figueiras 3 the obtained *OD* and $D_e$s are very similar considering the uncertainty (see Table 3). For Figueiras 2, the comparison is not possible as the EBG provides just five acceptable aliquots, therefore not statistically significant. However, the *OD* of the EBG for Figueiras 1 (FIGR-1 in Table 3) is very small ($2 \pm 1$%) when compared with the *OD* of the LBG ($20 \pm 4$), although both are small. The $D_e$ obtained with the EBG is around 23% smaller than these obtained with the LBG, but while the age obtained with the EBG is $23 \pm 4$ ka, these obtained with the LBG is $30 \pm 5$ ka. Similar ages are obtained for Figueiras 3 (FIGR-3 in Table 3) between EBG and LBG.

**Table 3.** Dose rate ($D_r$) (Grays/ka), equivalent dose ($D_e$) (Grays) and ages of the samples from the Cies Islands (FIGR: Figueiras 1, 2 and 3) (42°13′45″ N, 8°54′15″ W; WGS84), from EBG and LBG integration methods. Sampling distance (cm) from top to the base. (N) Number of aliquots accepted/analysed; (*OD*) Overdispersion percentage. (*) CAM-LBG ages (ka = kiloannum before dating) from [15]. Sampling distance (cm) measured from the top to the base of the outcrop.

| Samples | $D_r$ (Gy/ka) | N EBG | $D_e$ EBG (Gy) | OD-EBG (%) | Age (ka) (CAM-EBG) | N LBG | $D_e$ LBG (Gy) | OD-LBG (%) | Age (ka) (CAM-LBG) |
|---|---|---|---|---|---|---|---|---|---|
| FIGR-1 (200 cm) | $2.7 \pm 0.4$ | 24/96 | $62 \pm 2$ | $2 \pm 1$ | $23.3 \pm 4.2$ | 31/96 | $80 \pm 3$ | $20 \pm 4$ | $30.6 \pm 4.8$ * |
| FIGR-2 (140 cm) | $2.4 \pm 0.3$ | 5/54 | – | – | – | 36/54 | $56 \pm 5$ | $50 \pm 6$ | $23.3 \pm 3.9$ |
| FIGR-3 (80 cm) | $2.1 \pm 0.3$ | 26/51 | $38 \pm 2$ | $30 \pm 5$ | $18.9 \pm 3.8$ | 31/51 | $42 \pm 3$ | $33 \pm 5$ | $20.1 \pm 3.3$ * |

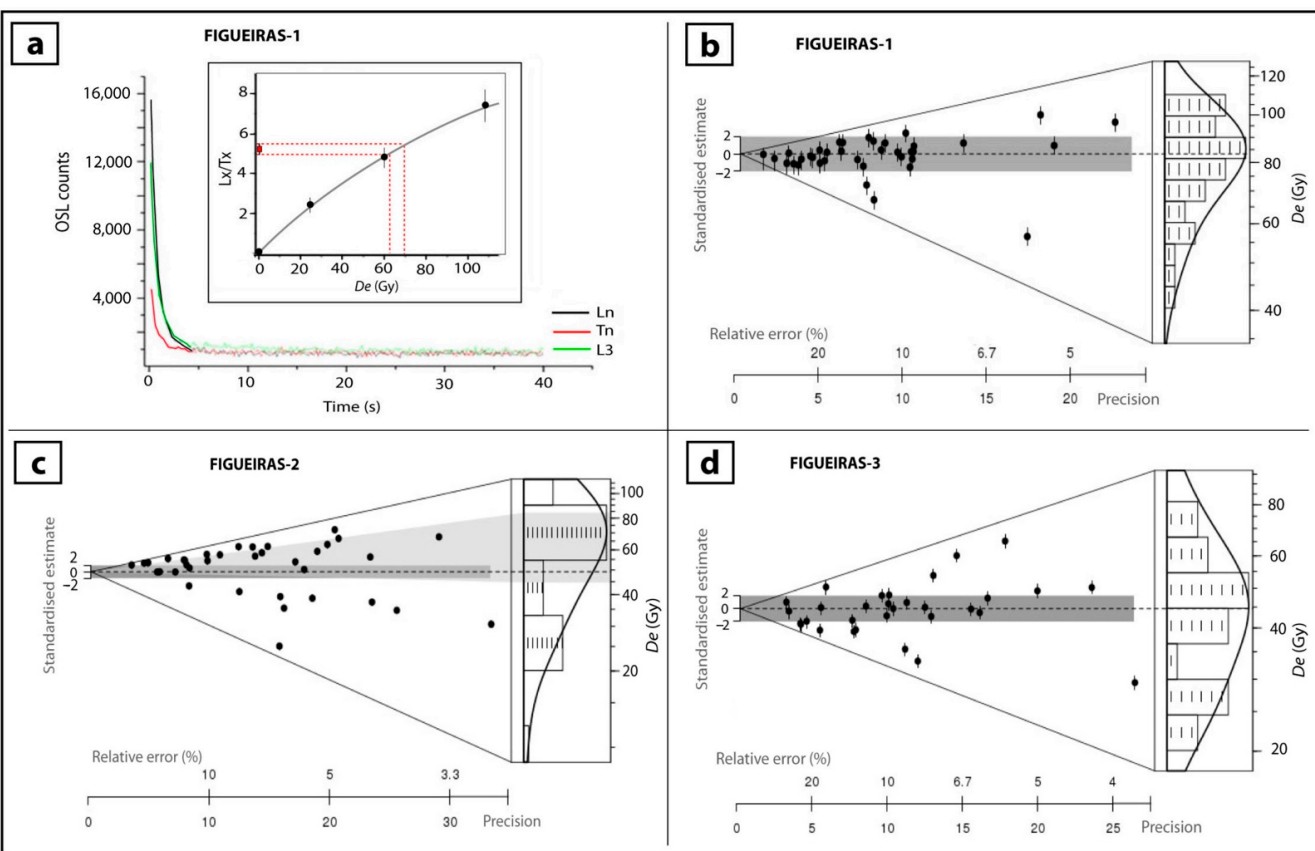

**Figure 6.** Quartz OSL signal and $D_e s$ distribution. (**a**) OSL-signal decay-curve from one aliquot of Figueiras 1 sample. Lines show the intensity (counts per second) from natural OSL-signal (Ln), natural OSL-signal from test-dose (Tn) and OSL-signal from third regenerative cycle from SAR protocol [44]. (Inset) Exp + lineal adjust for OSL growth-curve from normalised OSL-signal from SAR (Lx/Tx) versus equivalent dose ($D_e$), including the linear interpolation [58] from natural normalised OSL-signal. (**b**–**d**). Distribution of data from Figueiras 1, 2 and 3 samples, through Abanico plots [61]. Black dots represent $D_e$ estimates from accepted single aliquots, showing error bars. Grey bar in the radial plots represents the standard error ($\pm 2\sigma$) from the standardised estimate (weighted mean) by the Central-Age Model (CAM) (dotted line). Kernel-density estimate plot is also represented. Light-grey area in (**c**) represents the standard error ($\pm 2\sigma$) from the kernel density estimation.

Thus, there are not significant differences in the obtained age when both methods are used. Final ages from Figueiras 1, 2 and 3 (Table 3) are considered using the LBG method. These results are coherent with the ages obtained for other aeolian sands (aeolianites) along the Galician coast [16], which show similar behaviour of the OSL signals. All these ages are also consistent with other absolute ages calculated for fossil dunes in the coast of SW France, Portugal, SW Iberia, Gibraltar and the Canary Islands [22,62–67]. As discussed later, this clearly indicates a major episode of coastal aeolian accretion since the end of MIS2 in this Atlantic region.

## 5. Discussion

The evolution of the Atlantic coast of Galicia during the last regression (Upper Pleistocene) is poorly understood. The same happened with the stages of sea-level rising and flooding the Galician fluvial valleys (Rias) during the Holocene transgression. This is due to sediments of aeolian origin having been misinterpreted as marine deposits (see Section 1). Concerning Ria de Vigo, isolation of the old river valley from the sea waters 9 ky ago [30] and an incipient connection with marine waters from less than 3 ky was proposed [68]. In the innermost areas of this ria, a fossil forest 7.9 ky old was buried at −13 m (bpsl) (Table 1)

by a sand dune more than 10 m thick, indicating a lower sea-level at that time. The same occurs in the outermost areas of the ria where partially flooded dunes appear between 25–4 ky (Table 1)—along with 20 m thick dunes that accreted from more than 6 ky (Table 1), now flooded up to 35 m in depth. All of these sands indicate a lower sea-level that fits very well with the aeolianite outcrop preserved in the Cies Islands [15], older than 20 ky, correlated with a coastline lower than 100 m below present sea-level (bpsl) (Figure 7). Such data have allowed us to contextualise many of the isolated results available in references for the Ria de Vigo (Figure 4; Table 1) and the Galician coast, with the base of the reconstructed geomorphologic history for the coastal evolution from the end of the Upper Pleistocene to the present day (Figure 7).

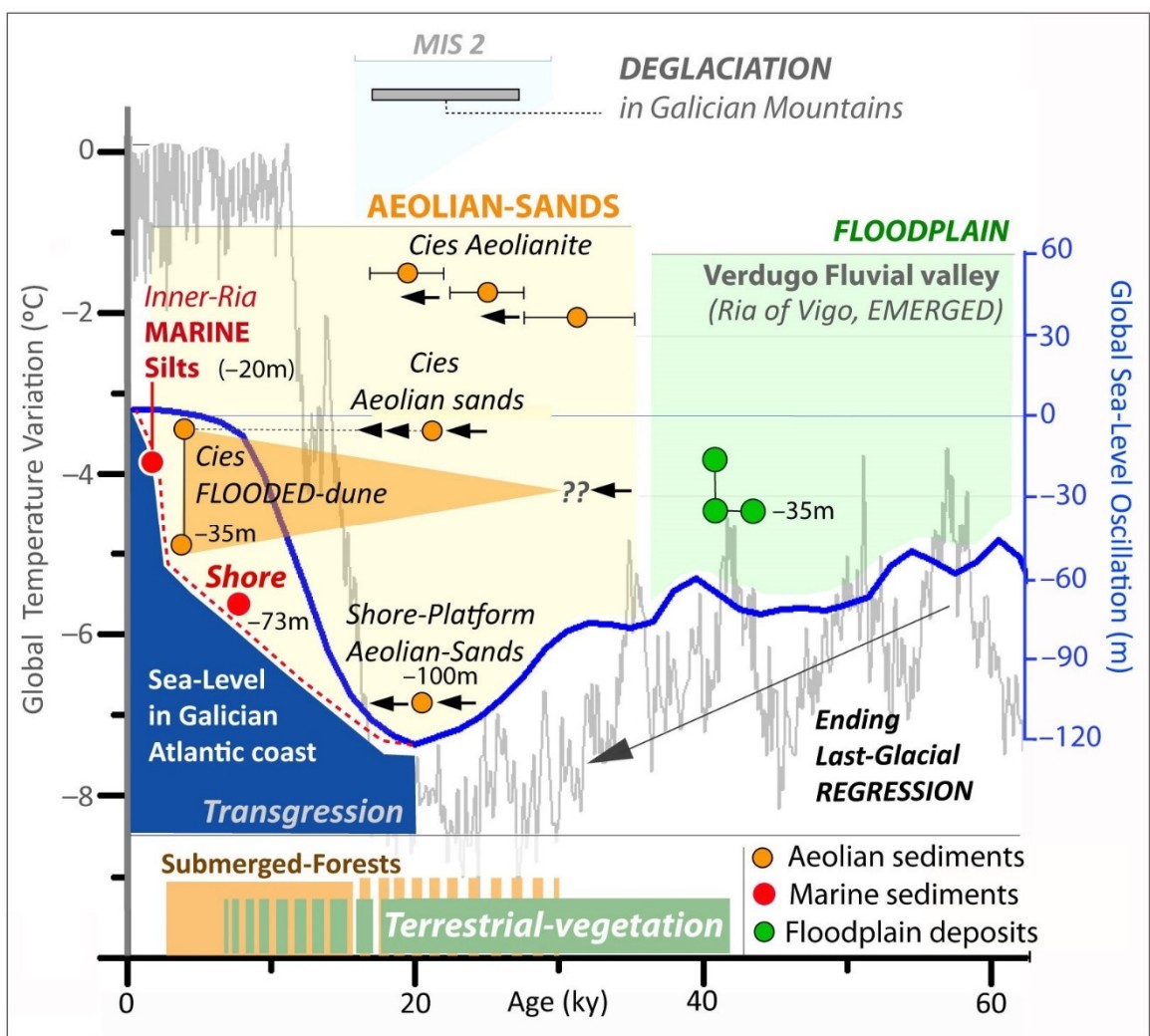

**Figure 7.** Local sedimentary record of the last 45 ky (continental and marine) in the Cies Islands and Ria de Vigo, compared to trends in the global climate record [69] indicated by temperature variations from Vostok ice cores [70] (grey line) and sea-level oscillations [71] (blue line), including the deglaciation processes in the mountains of Galicia and northern Portugal [72,73] at MIS2 [74]. Sedimentary record at the end of the Last Glacial period are represented by (i) floodplain deposits (green areas) and (ii) aeolian sands (yellow area) ([8,10,12,15] and this paper). Submerged forests buried by dunes from 30–3.5 ky along the coasts of Galicia and the northern Portugal [4]. Dark-blue area under the red dashed line shows the sea-level rise in the Ria de Vigo during post-glacial transgression, from the local record.

### 5.1. Aeolian Sediments in the Cies Islands and Ria de Vigo

The age of the aeolianites outcrop on Alto da Figueira (Isla de Monteagudo, Cies Islands) allowed us to establish an important aeolian accretion stage in the Atlantic margin of Galicia at the end of the Last Glacial period (MIS2) (Figure 7), coinciding with the maximum regressive levels in the northern hemisphere [71]. Additionally, within this aeolian accretion event there are other aeolianites preserved at various points along the Galician coast (Figure 2a,b) with age ranges of 20.9 ± 6.5 ky, 30.9 ± 3.6 ky and 29.7 ± 4.8 ky [16]. In the continental platform close to the Cies Islands, a 20 ky old deposit of aeolian sands [12] (Table 1) has been reported at −100 m (bpsl) (see record 1 in Figure 4; Figures 7 and 8a), which implies a lower sea-level at the end of MIS2 (Figures 7 and 8).

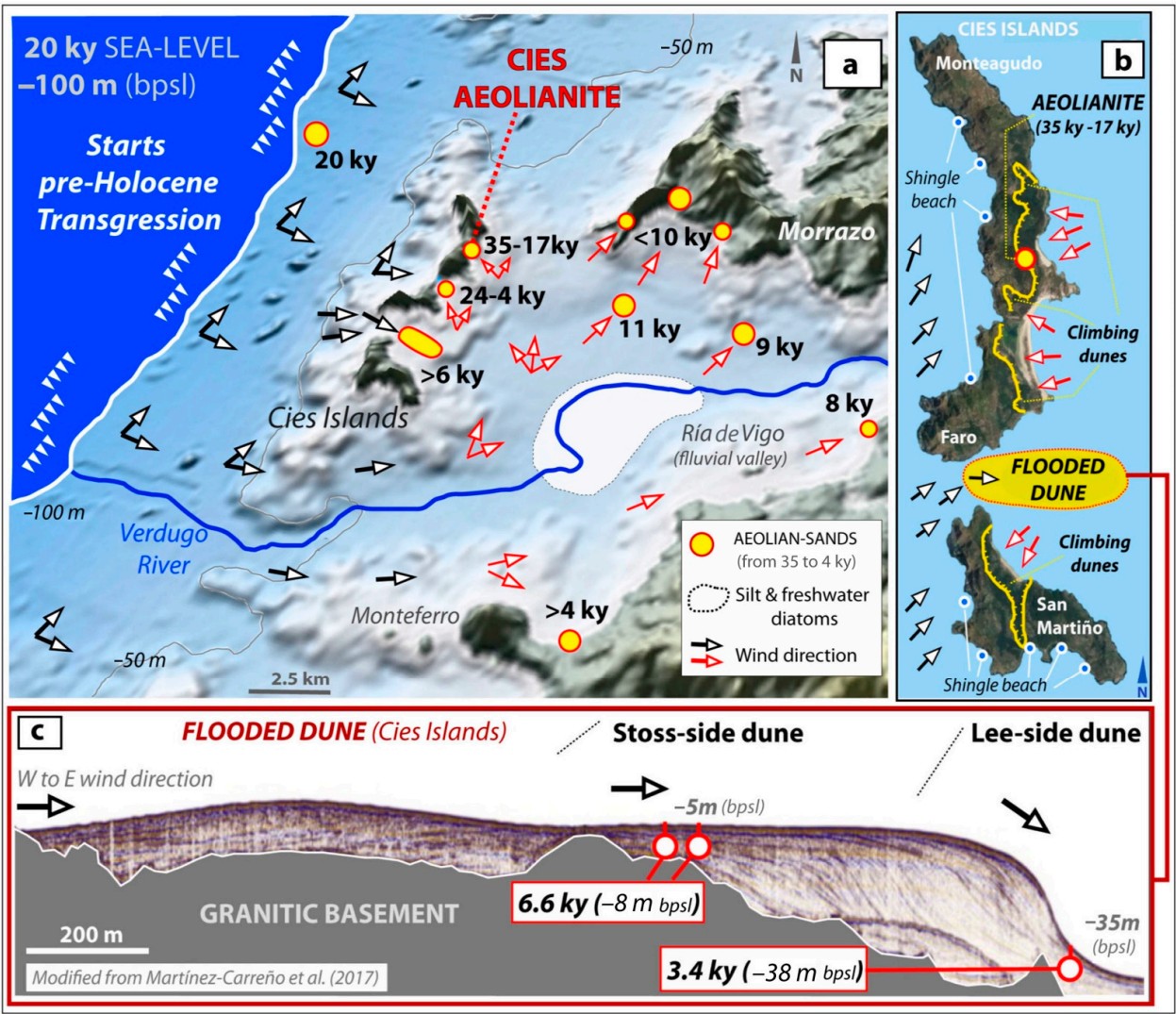

**Figure 8.** Aeolian sands in the Cies Islands, Ria de Vigo and continental platform. (**a**) Sea-level in the study area, showing the prevailing coastal winds from SW to NE (black arrows) in the Atlantic margin of Iberian Peninsula at the end of the Last Glacial period (modified from [15]). Red arrows show local wind variations, as suggested by the aeolian sands outcrops (yellow dots). The white area represents silty sediments with freshwater diatoms [27] in the uppermost levels, most probably related to freshwater diatoms in silty sediments described in [75] between 45 to 8 ky. Relief from [26]. (**b**) Prevailing wind direction in the Cies Islands at present sea-level and local variations suggests the climbing dunes advance (yellow stripes) as yet preserved in the archipelago [76]. (**c**) Flooded dune between Faro and San Martiño Islands, showing its internal structure [77] and chronology [28]. Black arrows show wind direction. (bpsl): below present sea-level.

At that time, glacial ablation processes were already taking place in the mountainous systems of the interior of Galicia (Figure 7) and northern Portugal [72,73]. This means, from a glacioeustatic perspective, that the Last Glacial maximum in the NW Iberian Peninsula must have been earlier, even reaching a regressive sea-level below −100 m (bpsl); similar features are also present in other part of the world [78,79]. Therefore, the beginning of this aeolisation began at the end of the last regressive stage, confirming the existence of an emerged sand strip more than 5 km wide in the continental platform of Galicia (Figure 8a), as the source of aeolian sand [15,21]. Considering that no other sources of aeolian sands are observed on this coast, the location of the fossil dune fields around Cies were built by sands transported by prevailing winds from SW to NE (Figure 8a,b) [15]. This regional wind dynamic has also been described to explain the evolution of the aeolian sands in the Atlantic coast of Portugal (Iberian Peninsula) at the end of MIS2, based on the sedimentary structures of dunes, OSL chronology and long-term regional climate models [22]. Furthermore, the bedding structures from the stoss-side and the lee-side of the now flooded dune in Cies (Figure 8c) agree with this regional palaeowind direction.

### 5.1.1. Aeolianite Outcrop in the Eastern Side of the Cies Islands at MIS2

Grain size, morphology and SEM analysis show that the sand deposit at Alto da Figueira (Monteagudo Island; Cies Islands; Figure 3) is of aeolian origin [5]. During the aeolian accretion that formed the aeolianite outcrop dated here, this fossil climbing dune advanced on the eastern side of Monteagudo Island (Figure 8a,b), reaching 3 m thickness at +40 m (apsl) [15]. This clearly indicates that the aeolian sands transported from the emerged shelf (Figures 4 and 8a) reached the interior of the Ria de Vigo, thus overcoming the topographic obstacle of the Cies Islands. Assuming a sea-level of −120 m (bpsl) at that time [71] (Figures 7 and 8), the three islands of Cies were connected to each other, and to the mainland (Figures 4 and 8a), reaching heights over 300 m above sea-level. This vigorous relief of Cies would be presumably a sedimentary trap for the aeolian sands mobilised from west to east [15] (Figure 8a,b), stopping the dunes from advancing and causing their concentration at the lowest levels. Nevertheless, similar to other relict climbing dunes in the cliff coast of Galicia [5] (Figure 9), also present on the northwest coast of the Ria de Vigo (records 5 and 6 in Figure 4; Figures 8a and 9c,d), the topography of the Cies Islands would not cause a major problem for the movement of aeolian sands that, in any case, could be channelled through the corridor between the islands of Faro and San Martiño (Figures 1, 3a, 4 and 8a,b), as explained below (Section 5.1.2). The growth of climbing dunes on the eastern slope of Cies (Figure 8b) implies a twist in the wind direction to drag the sands westwards [15] (Figure 8b). These local variations in wind movement are a common phenomenon even today as proven by the wind-faceted vegetation present on the islands (Figure 3b). All of these circumstances are also observed on nearby islands such as Ons Island [76] (Figure 1), located less than 30 km to the north (Figures 1 and 4). In this sense, more comprehensive studies of the ventifacts present on these islands are highly recommended.

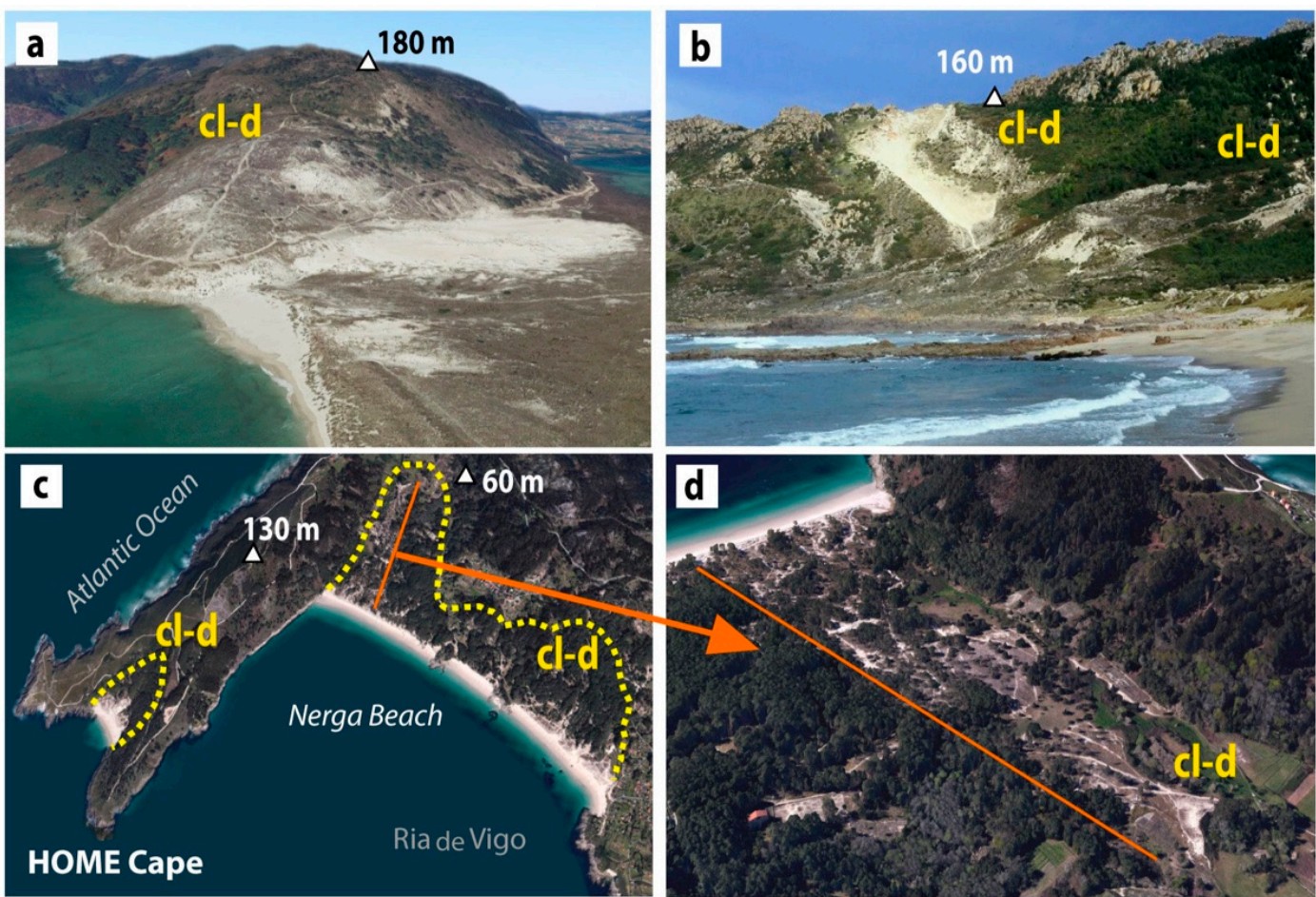

**Figure 9.** Relict climbing dunes still preserved in the Galician coast, reaching heights of more than 100 m. (**a**) Ponteceso (43°14′00″ N, 8°56′11″ W); (**b**) Trece (43°11′15″ N, 9°08′45″ W). (**c,d**) Relict climbing dunes near the Cíes Islands (northwest coast of the Ria de Vigo). (**c**) Home cape (42°15′07″ N, 8°51′57″ W); the yellow dotted line shows the climbing dune advance. (**d**) Nerga Beach and adjacent climbing dune (42°16′09″ N, 8°51′15″ W) (WGS84); cl-d: climbing dunes.

5.1.2. Flooded Dunes in the Cies Islands and Sea-Level

The sedimentary sands reported in the corridor between Faro and San Martiño Islands (Cies) (Figure 8c) were recognised by submarine geophysical campaigns [77], although they were misunderstood as an underwater sedimentary construction (never before described) formed by bottom currents during strong storms and where aeolian sands were confused with marine sands. Nevertheless, sedimentary features, absolute chronology and hypsometric data confirm that it is not a submarine formation but a subaerial one, as other studies described on other coasts of the world (see seismic profile of Figure 8 in [80]). Firstly, this sandy formation at 5 m depth has physical continuity with the dune fields of Rodas Beach, located at less than 1500 m between the islands of Monteagudo and Faro of Cies (Figure 1; aeolian sands in Figure 3a; record 3 in Figure 4; Figure 8a,b), which were formed between 20 and 4 ky [8] (Table 1). In this sense, another flooded sand deposit more than 13 ky old [14] and up to 20 m thickness was also described on the Areoso islet of the Ría de Arousa (Figure 1), close to the Ría de Vigo. The upper levels (2 m) of this deposit were accreted from 13 to 4 ky (see core A14-VC4 in [14]), as dune fields between the Cies Islands [8], abovementioned. In addition, the prolongation of these currently flooded sands culminates in a dune field whose upper level was dated by OSL at 2.5 ky [81]—covering both a 6 ky old megalithic tomb and a 4.6 ky old fossil forest. This reiteration of submerged sand bodies that are physically connected to emergent dunes would indicate that these are

large partially flooded dune fields, although only the upper levels that remain emerged are currently visible.

Secondly, the sedimentary structures of this flooded formation (Figure 8c) are related to a topographic-impeded dune close to lee-dune type [82,83], composed of fine to medium sands about 30 m thick. It clearly shows aeolian structure [84], with an extreme located stoss-side at −5 m (bpsl) (Figure 8c) whose formation culminated 6.6 ky ago (Table 1). Furthermore, these sands are extended in this corridor for over 1500 m in an easterly direction (towards the interior of the ria), where it is presently a well-developed lee-side reaching a depth of more than −35 m (bpsl) (Figure 8c). There, dune vegetation has been identified at 3 m below the deposit (−38 m bpsl) [28], dating back to 3.4 ky cal BP (Table 1). According to this, dune vegetation from 6 ky is also described in the nearby and now submerged dune fields of Rodas Beach [8] (Figure 3), reinforcing the idea of a continuum formed by aeolian sands. It should be noted that in this sand body, only the occasional presence of test is described in some of the few cores studied [28,77], without determining the species and community associations that define a marine environment—dissimilar from other studies carried out on the Galician coast [39,85]. Studies carried out in the Ria de Vigo have also failed to consider the wind mobilisation of the scarce test fragments found (e.g., from aeolian polishing marks and sorting by size), as recognised in previous studies on the Galician coast [39,85], thus dismissing the marine origin of these sands in favour of an aeolian formation.

Although this impeded dune is now completely flooded (Figure 8c), the reinterpretation proposed here is consistent with the data that places a transgressive sea-level at −73 m (bpsl) on the inner platform of Cies 9 ky ago cal BP [30]. Even more, its aeolian origin is really consistent with the data that place sea-level well below −13 m (bpsl), 7.9 ky ago cal BP (Table 1), as proven by the aforementioned submerged fossil forest of the Arenal (city of Vigo) in the inner section of the Ria de Vigo (record 9 in Figure 4), where whole fossil trunks and roots buried by a sand dune thicker than 10 m can be observed [29]. Later deposits indicate a well-developed continental ecosystem that was buried by those dune fields that reached the bottom of the ria and, at the same time, the aeolian accretion of the sand dunes with dune vegetation at the mouth of the ria (Cies Islands) was coming to an end. That was the case for the dune fields of Rodas Beach between the islands of Monteagudo and Faro [8] (Figure 3), and the now-flooded impeded-dune between the islands of Faro and San Martiño (Figure 8a,c). This clearly indicates that all of these areas were under subaerial conditions, in agreement with the disconnection from ocean waters suggested for this fluvial valley (the emerged ria) until 4 ky [30], as also suggested by test studies carried out in the Ría de Vigo at −40 m (bpsl) less than 3 ky ago [68]. Thus, the bathymetry of this flooded dune between the Faro and San Martiño Islands and its chronology should undoubtedly indicate a sea-level position lower than the stoss-side dune (−5 m bpsl) 6 ky ago (Figure 8c). Moreover, the sea-level would be lower than −38 m (bpsl) 3.4 ky ago on the basis of the chronological data [28], both the age and the depth of the uppermost level at lee-side dune (Figure 8c). This type of flooded aeolian formation can also be observed on the nearby Ons Islands (see the same transect PT7 in [86–88]) and Sálvora Island (see Figure 13c in [11]), developing a similar topographic and sedimentary structure—although they have not been mentioned until now. These sands would later be transported by the wind, covering the eastern leeward side of these islands (more sheltered from the strong coastal winds) and also reaching the current coastline of the Ria de Vigo (Figure 10), as will be explained later. According to the model proposed here, these data would appropriately fit the sea-level rise curve for the western Galician Rias during the Early–Middle Holocene, as represented in Figures 7 and 10.

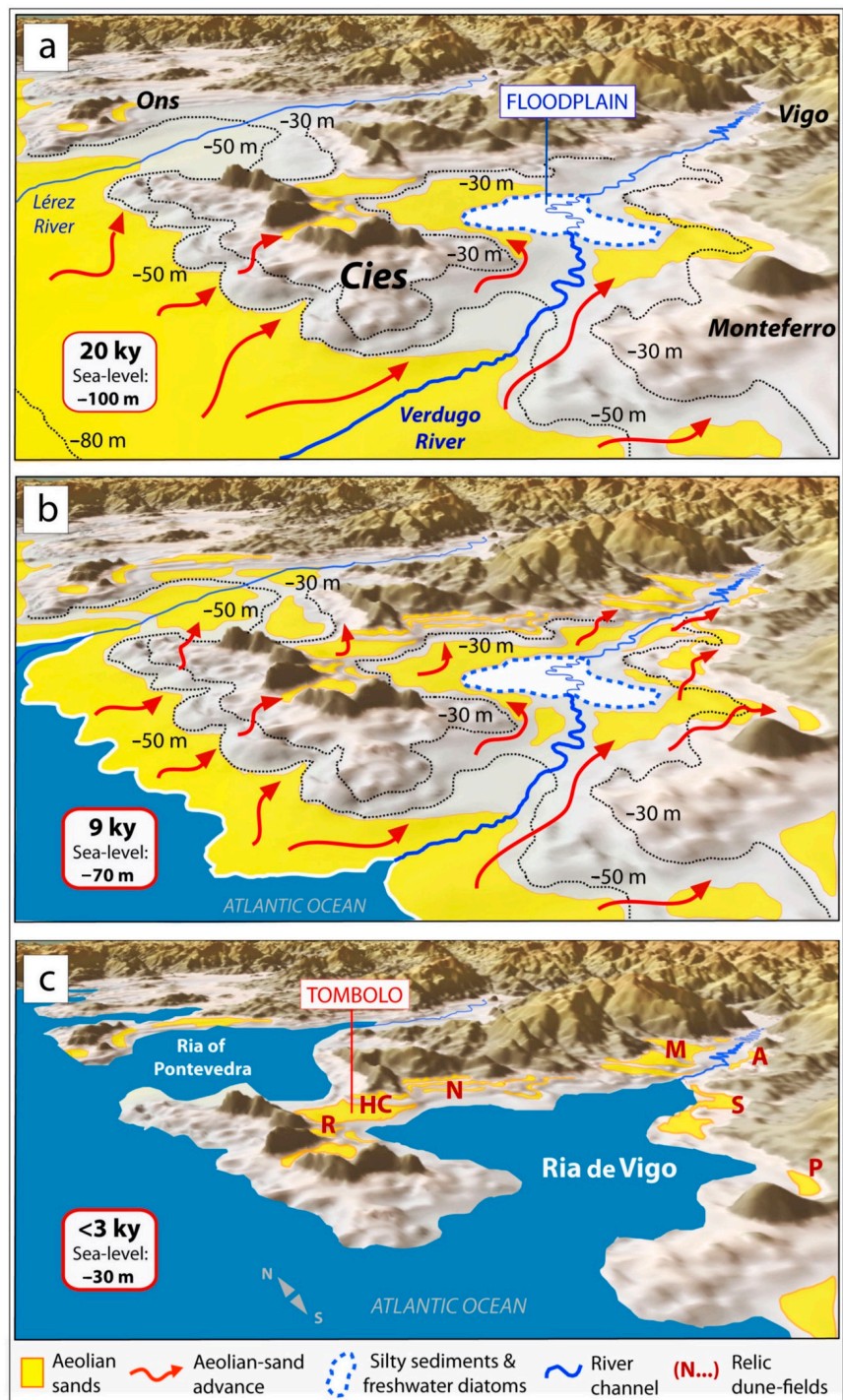

**Figure 10.** Sequence of the aeolian accretion process in the fluvial valley of Verdugo River at the end of the Last Glacial period, showing the hypothetical sand dunes advance towards the continent (red arrows) and subsequently sea-level rise in the last post-glacial transgression: (**a**) 20 ky aeolian sands advanced through the emerged continental platform and the Cies Islands from a sea-level below −100 m (bpsl); (**b**) 9 ky sea-level about −70 m (bpsl), with the aeolian sands upwards within the fluvial valley; (**c**) incipient configuration of the Ria de Vigo with a sea-level about −30 m (bpsl) from less than 3 ky, flooding the aeolian sands and the river plain. Capital letters indicate aeolian sands that are related to the relict aeolian formations (as dune fields and climbing dunes) currently preserved around the ria: Rodas (R), Home Cape (HC), Nerga (N), Moaña (M), Arenal-Vigo (A), Samil (S) and Patos (P).

5.1.3. Dunes Filling the River Valley and Environmental Collapse

Mobilisation of dune fields towards the continent remains active as the sea-level rises during the Holocene transgression [4], although the aeolian supply was reduced as the source area was flooded by the rising sea-level. Considering a sea-level at −73 m (bpsl) 9.4 ky ago [30] and below −38 m (bpsl) 3.5 ky ago (Figure 10), as suggested by the chronology and bathymetry of the flooded dune of Figure 6c, the relief of Cies at that stage was still emerged and joined the Peninsula de Morrazo through another sand [7] tombolo (Figures 8a and 10). In general, a marine origin has been assigned to the sand deposits around the tombolo, as shown by the models of coastal evolution previously proposed [8,10,86,88]. Nevertheless, in the cores sampled at −35 m (bpsl) in the same area (see cores VC2 and MRV3 in [10] and records 7 an 8 in Figure 4), a deposit with a 2 m thickness of fine sand and silty sand (upper levels) are reported having an age of 11 ky cal BP at the base (Table 1). These authors assigned a marine origin to these sands based on the sporadic presence of possibly marine bioclasts, without considering the possibility of artefacts typical induced by the same vibrocorer sampling, quite frequent and also mentioned in other studies in Galicia [5,38,39,85]. However, the chronological and bathymetric data analysed and discussed here show that these 11 ky sands located at −35 m (bpsl) (Figures 8a and 10) can only be subaerial and therefore aeolian sands. It is only under this context that we could favourably justify their physical connection with the Holocene aeolian deposits [8] located on the northwestern coast of the Ria de Vigo (Peninsula de Morrazo), such as the relict dune fields and climbing dunes of Nerga beach and the cape Cabo Home (Figures 1, 4, 8a, 9c,d and 10). Once again, a continuum is observed, as transgressive aeolian sand sheets accreted during the Holocene from a lower sea-level (Figure 8).

The coastal aeolian accretion described here also triggered the ecological collapse of the continental environments produced by all those aeolian sands that were able to reach the innermost areas of this valley (now the present-day Ria de Vigo) (Figures 4 and 10). For instance, in the sedimentary record described at −35 m (bpsl) in cores VC2 and MRV3 [10], lower strata of silty layers with woody remains from 45 to 40 ky cal BP (Table 1) were identified (see floodplain deposits in records 7 and 8 in Figure 4; Figures 8a and 10). These materials, similar to other silty sediments described in the inner ria with the presence of freshwater diatoms [75], are related to the fluvial valley (Verdugo River) when sea-level was below −100 m, and later covered by aeolian sands dated back 11 ky. It should be noted that the descriptions of these cores (MRV3 or VC2) [9,10] also differ, with no presence of foraminifera in [9]; in contrast, [10] mentions a sporadic presence of foraminiferal tests that are unable to establish a marine origin for these sediments, contrary to what the authors suggest. Thus, these currently flooded sands were surely accreted by wind, although the advance of dunes was stopped by the river course, as proven by the contact between the aeolian sands and the finest sediments [12] of fluvial origin (Figures 4, 8a and 10). Additionally, in this muddy area with silts and clay silts, an association of freshwater diatoms were also identified [27] along the path followed by the river (Oitavén–Verdugo River), as suggested by other authors based on hypsometric data (see Figure 3 in [10]). This silty area with a low slope [26] is interpreted here as a floodplain with a meandering course (Figures 5 and 10) and the lack of aeolian sands on it would be positively justified by the fluvial dynamics that prevented the accretion of the windblown sands.

As to the southern coast of the Ria de Vigo, the accretion of aeolian sands up the valley (Figures 8a and 10) is also represented by a 10 m thick sand dune that buried the forest of Vigo–Arenal (see record 9 in Figures 4 and 11a–c), located at −13 m (bpsl) and dated back 7.9 ky [29] (Table 1). This dune preserved in the bottom of the ria is spatially and temporally related to 2 m thick flooded sandy sediments, described at −20 m (bpsl) (see MRV4 and B5 cores in [9,10]), which dated back 8.8 ky cal BP (Table 1) (see sequences 9 and 10 in Figure 4). Although, once again, a marine origin was assumed by these authors (cores MRV3 and VC2) based mainly in global correlations; these sand sediments are also reinterpreted in our work as aeolian sands due to their absolute ages and depth, considering their physical

connection with nearby sand deposits from aeolian origin (Figures 4, 5 and 10). These aeolian sands entered the ria on the southern side of the mouth of the Oitavén–Verdugo River and Monteferro promontory (Figure 10). This would be the case for the dune fields related to Samil Beach (Vigo) or Patos Beach (Nigrán) (Figures 1 and 10), where remains of fossil wood (in living position) have now appeared within the intertidal zone, dated back 4.6 ky [30] (sequence 11 in Figure 4; Figure 11d,e). Other forest soil with roots and trunks in living position (Figure 11d,e), were described along the Galician coast between 8 and 3.5 ky [4]. Thus, the woody remains and arboreal pollen under the 20 ky old dune fields in the Cies Islands [8] and all these ancient forests were buried by aeolian sands. This meant the ecological collapse of coastal environments by dunes from 25 to 4.5 ky, which preceded the marine flooding of the valley, as a major environmental crisis caused by climate change and global warming during the Holocene.

All chronological and sedimentological evidence allow us to relate these continental deposits to the remaining submerged forests buried by sand dunes that have been described along the Galician and northern Portuguese coastline [89–91]. These were well-developed forest formations with deciduous species growing away from marine influence [4]. In turn, all of these continental formations affected by coastal aeolian sands, as proposed here, are related to submerged forests that appeared in the west of France and all around the British coasts at the same time [92–95]. Therefore, this aeolisation stage could be extended to the whole southern half of the European Atlantic coast, also suggesting that the absolute ages of the aeolianites preserved in SW France [67,96,97] together with SW Iberia, Gibraltar and Canary Islands [22,62,68,69] are from MIS2 to Late Holocene. Based on the chronology established for other coastal aeolianite outcrops in Galicia [21], the model can also be applied to the penultimate glacial episode and the Eemian transgression, and could also be extended to other coastal areas worldwide where coastal aeolianites have been dated [80,98–105].

### 5.2. Present-Day Sea-Level and Lack of Aeolian Supply

As mentioned, the fluvial valley of Ria de Vigo was disconnected from ocean waters until less than 4 ky, at which time benthic test associations are described in silty surface sediments in the Ria de Vigo at −40 m (bpsl) [68]. These data correspond to the final stage of the lowermost areas of this valley as the ocean waters reached the present-day levels, leading to the configuration of this wide estuary as it is today. This agrees with the marine silty sediments described at −20 m (bpsl), in the inner section of the Ria de Vigo (see MRV4 and B5 cores in [9,10]) dated back to 2 ky cal BP (Table 1), overlying older aeolian sands (record 10 in Figure 4). From that moment onwards, sea-level rise completely stopped the aeolian supply described above and initiated an erosive aeolian phase [5] that affected the emerged dune located on the present-day shoreline. Paradoxically, sea-level rise has favoured the preservation of a large part of the old aeolian formations within the Ria de Vigo (such as the flooded dune between the Cies Islands), probably due to the Cies relief hindering erosion by wave action at the mouth of the estuary, unlike the processes of wave washing and destruction of Early and Middle Holocene sand-dunes suggested in other types of coasts characterised by the presence of large islands as a natural barrier [106]. Subsequent flooding and/or marine reworking of aeolian sands would favourably explain the presence of salt crusts and polished surfaces on the sand grains [35], along with the presence of neoformation marine minerals, such as glauconite, associated with shallow environments [77].

Today, as sea-level rises, the waves and tides sweep away the coastal dunes forming the current sand beaches, as in the case for the Rodas, Nerga, Moaña, Arenal-Vigo, Samil and Patos beaches (Figures 1 and 10). As mentioned, this suggests that the sand beaches in this area are aeolian sands reworked by the sea. It means that the relict dunes in this coast are not simply an extension of sand beaches inland [107] (p. 11). On the contrary, they are sediments formed from coastal aeolian accretion from a lower sea-level that began at the end of MIS2 and continued into the Late Holocene.

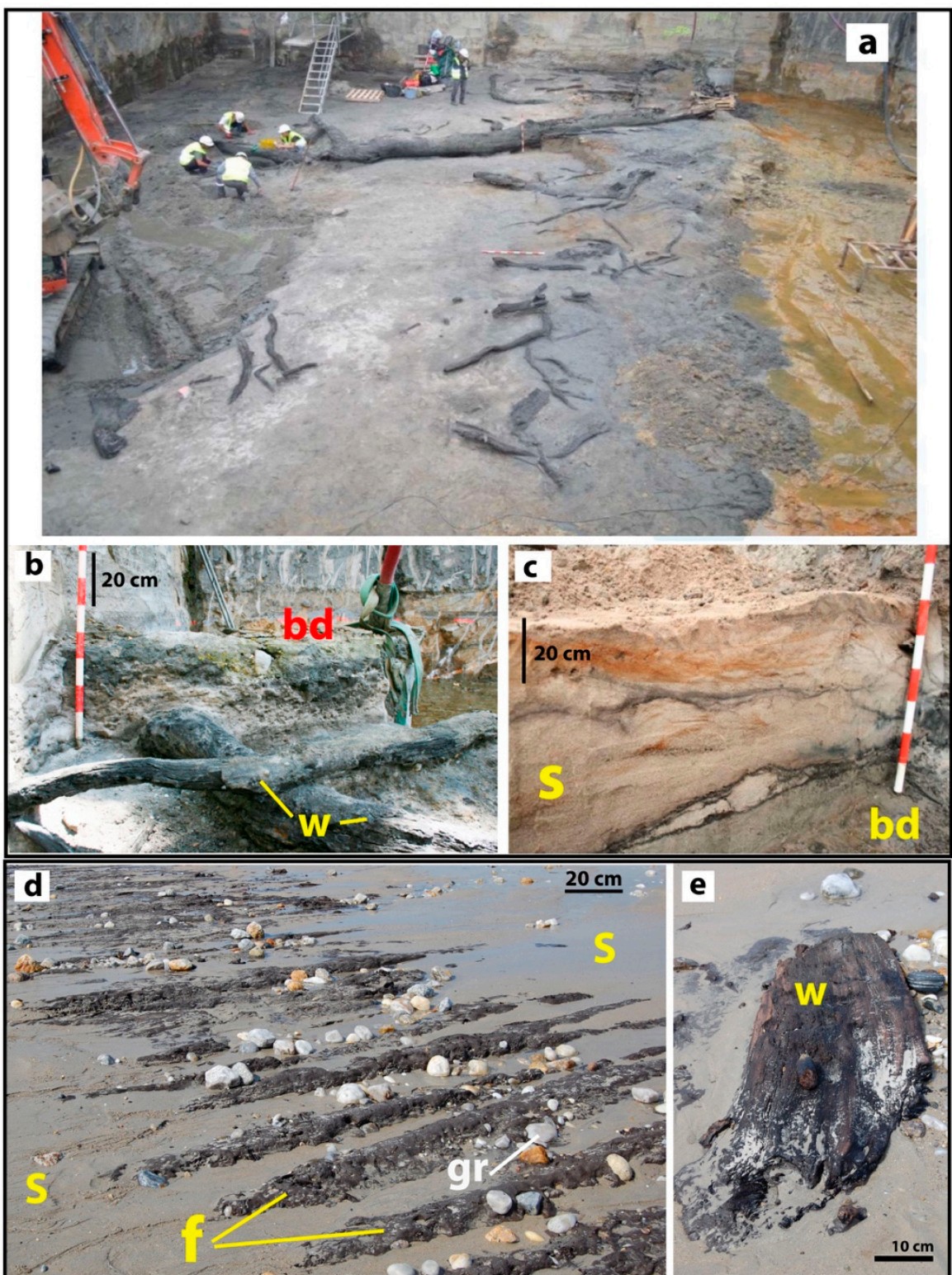

**Figure 11.** Submerged forests buried by Holocene dunes in the Ria de Vigo. (**a**–**c**) Submerged forest of Arenal (Vigo, Pontevedra) located at −13 m (bpsl) under a dune of over 10 m. (**b**) W: woody remains and whole trunks; bd: base of dune. (**c**) Detail of the aeolian sands from the dune of Arenal; S: aeolian sands (modified from [29]). (**d**,**e**) Submerged forest in Patos Beach (Nigran, Pontevedra); f: forest soil with roots in living position; gr: gravels; s: sand beach; w: wood remains.

## 6. Conclusions

The chronology assessed for the aeolianite outcrop of Monteagudo Island (Cies Islands) allows us to define a prolonged sequence of aeolian sedimentation between 35,000 and 17,000 years before present. Considering the age of this and other climbing dunes developed in the Atlantic margin of Galicia, our data suggest an episode of aeolian accretion coinciding with the end of the Last Glacial period (MIS2) with a massive mobilisation of aeolian sands towards the Cies Islands and the current coast of the Ria de Vigo (emerged during the Last Glacial period). Based on the correlation with the global palaeoclimatic curves, the dating of subaerial sand sediments in the continental platform of Galicia and the age of the aeolian deposits, we propose the existence of large areas of the continental platform (>5 km) completely emerged during the Last Glacial period.

The thickness of the Alto da Figueira aeolianite in Monteagudo Island (Cies Islands) and its location on the eastern slope (the most protected) suggests that during the last stage of the Upper Pleistocene the archipelago was emerged, connected with the continent and partially covered by aeolian sands. During this period (MIS2), the presence of active fluvial courses would prevent the advance of the dunes through the southern extent of Cies. Along the western slope, the aeolian processes would be constrained against abrupt vertical relief, being channelled by corridors such as those existing between the actual islands of the Cies archipelago (Monteagudo, Faro and San Martiño). In all of these areas, dunes that are totally or partially flooded are still preserved today. As to the north extreme of Cies, the connection with the continent would also be constituted by aeolian sands (as a tombolo), facilitating the development of Holocene dune fields and climbing dunes, such as those currently preserved on the northwesternmost coast of the Ria de Vigo (Peninsula de Morrazo).

During the Last Glacial period, oceanic waters completely abandoned the Ria de Vigo, reactivating the fluvial dynamics of the Oitavén–Verdugo fluvial system in this emerged valley. This main riverbed was developed over a wide floodplain for 50 km (approx.) covered by marine and continental sediments. On these emerged materials and throughout the Upper Pleistocene, the expansion of forests occurred along the emerged valley, present even in Cies. These well-developed forests were dramatically affected by the advance of the dune fields into the Ria de Vigo (then emerged) from 30 ky to less than 5 ky. This progressive aeolisation can be extrapolated to the entire coast of Galicia and northern Portugal, and most probably the southern half of the European Atlantic margin.

Based on chronological and hypsometric data available in the local sedimentary record, a sea-level rise curve is fitted for the Ria de Vigo, reaching less than −100 m (bpsl) at the end of the Last Glacial period (20 ky), −70 m (bpsl) in the Early Holocene (9 ky), −50 m to −40 m (bpsl) in the Late Holocene (3 ky) and −20 m (bpsl) to 0 m from 2 ky. Thus, the sea-level rise was fast enough (from 7 to 12 mm/y) to preserve all flooded aeolian formations (dunes) around the Cies Islands. This aeolian accretion stopped completely as the sea reached its present level and was replaced by wind and marine erosion.

**Author Contributions:** Conceptualisation, C.A.-C. and J.R.V.-R.; funding acquisition, J.R.V.-R.; methodology, C.A.-C. and J.S.-S.; investigation and resources, C.A.-C. and J.R.V.-R.; supervision, J.R.V.-R.; writing and reviewing, C.A.-C. and J.R.V.-R. All authors have read and agreed to the published version of the manuscript.

**Funding:** This research was funded and supported by Xunta de Galicia (programmes ED431B 2018/47 and ED431B 2021/17) through the Grupo Interdisciplinar de Patrimonio Cultural e Xeolóxico (CULXEO).

**Institutional Review Board Statement:** Not applicable.

**Informed Consent Statement:** Informed consent was obtained from all subjects involved in the study.

**Data Availability Statement:** The data used in this study were obtained from the doctoral thesis of the first author (C.A.-C.) (https://ruc.udc.es/dspace/handle/2183/19810) (accessed on 7 August 2022).

**Acknowledgments:** We would like to thank José Antonio Fernández Bouzas, director of the PNMTIIAA-Islas Cíes, for his permission to visit the park and take samples. We would like to thank the Xunta de Galicia for its support through the Grupo Interdisciplinar de Patrimonio Cultural e Xeolóxico (CULXEO) (programmes ED431B 2018/47 and ED431B 2021/17). We would also like to thank most sincerely the Reviewers and the Editors for their suggestions and comments.

**Conflicts of Interest:** The authors declare no conflict of interest. The funders had no role in the design of the study, in the collection, analyses or interpretation of data, in the writing of the manuscript or in the decision to publish the results.

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
