# Peer review of "New Model of Coastal Evolution in the Ria de Vigo (NW Spain) from MIS2 to Present Day Based on the Aeolian Sedimentary Record"

_jmse, doi:10.3390/jmse10101350_

Round 1

Reviewer 1 Report

Sea level changes along the Atlantic coasts are of academic significance in understanding global climate changes and its impacts on human activities. The work presents an integrated study of geomorphology and geochronology to reveal regional sea-level influence since the last glacial maximum, and is thus worthy of publishing in JMSE. However, because of the comments listed belowe, it is hard to recommend its acceptance at present.

1) This is not a review paper, why the part of introduction and background is so long, till Line 234? If it is a paper between review and regular, they should be put in discussion and supplementary materials. 

2) Are those of methods and results not important to assess the reliability of the study?

3) I guess Fig. 8 was from a combination analysis of a series seismic profiles and borehole studies, and I think these information should be mentioned in details in supplementary materials

4) I am not sure sections 4.1.3 and 4.2 are necessary in discussion. If the authors want to say that their model is applicable in other periods, why not show more results, and if only to show the potential, why so many words in these sections?

Thus, I think the work is interested but fails to point their findings and significance clearly.

Author Response

Reviewer 1

Sea level changes along the Atlantic coasts are of academic significance in understanding global climate changes and its impacts on human activities. The work presents an integrated study of geomorphology and geochronology to reveal regional sea-level influence since the last glacial maximum, and is thus worthy of publishing in JMSE. However, because of the comments listed belowe, it is hard to recommend its acceptance at present.

1) This is not a review paper, why the part of introduction and background is so long, till Line 234? If it is a paper between review and regular, they should be put in discussion and supplementary materials. 

REPLY: Thanks for the comment of Reviewer 1. This paper is an article that reviews previous data and its reinterpretation according to the new contributions on the evolution of the coastline. In this case the dating of the Cíes Islands eolianite in the Vigo estuary carried out by the authors. The age of this fossil climbing dune from the Cíes Islands at +40m (apsl), between 34 and 15 ky in age, prompted us to carry out an exhaustive review of the aeolian deposits in the whole area (Ria de Vigo and Cies Islands) but also in the rest of the Atlantic coast of Spain and in other areas of the world as a good example of coastal evolution from the end of the last Glacial Period to the present day. We agree with Reviewer 1 that the Introduction and, above all, section 2 (State of the Art) is long, but there is a relevant reason. It was precisely here that the type of Rias coast was defined (Richthofen, F. v. (1901): Fiihrer fiir Forschungsreisende (Neudruck der Aufl. von 1886), Hannover). On the basis of new knowledge, it is possible to correct the errors introduced more than a century ago and which have been passively accepted by all authors. To highlight the role of aeolisation from the end of the last Glacial Period to the present day in the accumulation of the sediments now visible on the coast is of great importance not only for the Spanish coast but also for the rest of the world which has been affected by the same type of sedimentary processes. These aspects have been ignored in the previous literature despite the fact that these previous works provided valid sedimentary and even chronological data. All the data covered in the Discussion have been presented in the previous sections, for a better understanding of the Discussion. However, as suggested by Reviewer 1, we have reduced the length of these sections in order to improve the text. Aspects presented in the Introduction above become part of the discussion. Section 2.1. becomes the introduction (as suggested by Reviewer 2). Supplementary material has also been removed and Material and Methods and Results have been included in the main text.

2) Are those of methods and results not important to assess the reliability of the study?

REPLY: Thanks for the comment of Reviewer 1. As mentioned above, the main text has been corrected and the headings Material and Methods (4) and Results (5) have been included in the main text, as requested by Reviewer 1.

3) I guess Fig. 8 was from a combination analysis of a series seismic profiles and borehole studies, and I think these information should be mentioned in details in supplementary materials

REPLY: Thanks for the comment of Reviewer 1. Figure 8 is a (hypothetical) model to explain the evolution and current location of the existing aeolian deposits in this specific area (Cíes Islands and Vigo estuary), which coincides with those observed in nearby areas, that is, along the entire Atlantic coast of Galicia (NW Spain). All of them are described throughout the text (boreholes, seismic profiles, subaerial deposits, bathymetry and dating), citing the appropriate references for consultation. As mentioned above, all these data have been interpreted in a confusing or even erroneous way. However, if all these data are contextualised, as we have tried to achieve in a coherent and orderly manner, a logical hypothesis can be established to interpret the evolution of the coast from the end of the last glaciation to the present day. This has allowed us to advance our knowledge of the evolution of the coast in this area since the end of MIS2. This is the main achievement of the work.

4) I am not sure sections 4.1.3 and 4.2 are necessary in discussion. If the authors want to say that their model is applicable in other periods, why not show more results, and if only to show the potential, why so many words in these sections?

REPLY: Thanks for the comment of Reviewer 1. The authors consider that section 4.1.3 (now 6.1.3) is extremely important to understand that the sands that today cover the bottom of the Ria de Vigo are of aeolian origin, and that they have been formed as a result, first, of aeolian accretion by dune fields and then flooded by the Holocene transgression. And all this can only happen with a lower sea level. And it can be extrapolated to the rest of the Galician Rias. This stage in the geological history of the Galician coast meant the collapse of coastal ecosystems, as can also be observed along the Atlantic coast of NW Spain, and in other parts of the world. Therefore, this is not a local phenomenon, but a generalised one that has induced important geomorphological and environmental effects, including the formation of the rias, which are not due to marine erosion but to the flooding of a primary coast (Ottman, F.C. Introducción a la geología marina y litoral. Editorial EUDEBA, Buenos Aires, Argentina, 1967; 287p.). A coherent interpretation of aeolian sediments and other coastal continental formations is essential to understand how and when they were emplaced on the Galician coast during the Holocene.

Section 4.2 (now 6.2) is also of great importance as it explains the end of the aeolian accretion in this coast, once the sea reaches its present level and occupies the entire source area of wind input. In addition, specific and novel chronological data are provided on the time when the sea flooded these valleys and shaped the Ria de Vigo (and therefore, the Galician Rias), data that are explained within a coherent context. All this has allowed us to construct a model of coastal evolution very different from the models described above, which place the formation of the Ria de Vigo and the Galician Rias more than 7 ky ago (Early Holocene) without considering at any time the episode of mobilisation of large dune fields that preceded the rise in sea level. Another important aspect of this section is that it explains the existence of coastal sandbanks (sandy beaches), as a consequence of the flooding of dunes by the sea, something that has not been previously described in the literature, at least in the study area. All this is described in the most synthetic way, i.e. without useless information or unnecessary fillers. Only information useful for understanding the hypothesis being put forward is included. We kindly hope that we have removed the doubts of Reviewer 1 in this respect.

Thus, I think the work is interested but fails to point their findings and significance clearly.

REPLY: Thanks for the comment of Reviewer 1. The main conclusion is that there was a large episode of wind accretion in the area since the end of MIS2 that preceded the sea level rise (e.g.: climbing dunes of more than 20 ky, between 10 and 5 km away from the coastline, then at -120 m (bpsl). This phenomenon has been ignored in previous works, which describe the flooding stage of the river valleys forming the Galician Rias (and specifically the Ria de Vigo), ignoring the previous stage of aeolian accretion, which is key to understanding the existence of these aeolian deposits on the surface and/or under the sea, as well as the coastal sand-beaches. This model explains how and why the dunes were introduced into the valleys burying the forests, which led to a dramatic environmental crisis that preceded the sea-level rise. The dating also leads us to consider a new (hypothetical) model of sea level rise (this is the hypothesis established on the basis of the data), very different from previous models, which is of great geomorphological importance.

With this, we believe to dissipate the doubts about the importance and conclusions of this paper, which presents a novel environmental approach in a coherent climate context. Finally, we would like to mention that the original Spanish text was translated by a native English philologist from the Department of English Philology of the University of Coruña. Subsequently, the authors made the relevant modifications in relation to geological, geomorphological and dating aspects. However, further modifications have been made to improve the comprehension of the text.

Reviewer 2 Report

This paper is very interesting and deserves to be published after minor revisions. The study deals with the coastal evolution in Cies Islands from the end of the Last Glacial period to the present day. Some important results have been reached, such as the detection of aeolian deposits in different areas of the Ria de Vigo.

I have some comments on the general structure of the manuscript. The introduction is mainly focused on the study area; in my opinion it is better to highlight the study of the rias and incised valley related to sea-level changes from the Last Interglacial to the Holocene, highlighting the methodology used in this work. Furthermore, the section Results must be inserted in the main text. There are some important results that deserve to be inserted in the main text, such as Quartz OSL signal and Des distribution. This helps the readers to better follow the Discussion.

In the Table 1, is it possible to insert the delta 13 for each sample if available? It is important for the data calibration.

Other comments have been inserted in the pdf attached file.

Author Response

Reviewer 2

This paper is very interesting and deserves to be published after minor revisions. The study deals with the coastal evolution in Cies Islands from the end of the Last Glacial period to the present day. Some important results have been reached, such as the detection of aeolian deposits in different areas of the Ria de Vigo. I have some comments on the general structure of the manuscript.

1- The introduction is mainly focused on the study area; in my opinion it is better to highlight the study of the rias and incised valley related to sea-level changes from the Last Interglacial to the Holocene, highlighting the methodology used in this work.

REPLY: Thanks for the comment of Reviewer 2. As suggested, the Introduction has been modified. The new introduction highlights the study of the rias and the incised valley related to sea level changes from the Last Interglacial to the Holocene, as well as the methodology used.

2- Furthermore, the section Results must be inserted in the main text. There are some important results that deserve to be inserted in the main text, such as Quartz OSL signal and Des distribution. This helps the readers to better follow the Discussion.

REPLY: Thanks for the comment of Reviewer 2. As suggested, the main text has been modified to include the sections 3. Material and Methods and 4. Results

3- In the Table 1, is it possible to insert the delta 13 for each sample if available? It is important for the data calibration.

REPLY: Thanks for the comment of Reviewer 2. In most of the reference works this data is not provided and we only have the uncalibrated ages (and those calibrated with INTAL13 and even earlier curves). The re-calibration of these data with INTCAL20 is only to homogenise data with the most modern version. However, it is important to note that the previous data and those recalibrated with INTCAL20 do not vary significantly, and have the same meaning. Furthermore, the absence of this parameter when using the OXCALL program (for the re-calibration with INTCAL20) would not invalidate these results.

Other comments have been inserted in the pdf attached file.

REPLY: Many thanks for the helpful comments of Reviewer 2. All suggestions made by Reviewer 2 have been considered and suggested modifications have been made to the text. The suggested references have also been included and the suggested errors of vocabulary and form have been modified.

Reviewer 3 Report

According to the authors, the main contribution of the paper is to explain the evolution of a sector of the rocky coast of Galicia, Spain (“Ria de Vigo”) since the last glacial period to nowadays, based on new data that are not well understood how they were obtained.

Although presenting an interesting subject the manuscript seems to be a poorly achieved synthesis of obtained data by the authors.

I do not think the manuscript is suitable for publication in its present form and needs major revision. Below I list some aspects that are on the basis of my decision. Authors will also be able to access the observations made up to page 8, on the submitted version.

- In general, the structure of the manuscript is rather confusing and consequently deserves a certain degree of reformulation.

- The title and abstract reflect the content of the article however the key words need to
be reviewed and be more objective.

- The introduction should include the section 2. State of the question (why this name?) because it is here where the authors explores comprehensively the available literature (state of art) although the objectives in mind are not very clear.

- Study area. This section needs to be improved regarding English form and grammar. Some geological and physical processes terms should also be reviewed.

- The paper does not present in the body of the manuscript the sections related to methods and results. In a scientific paper the methodology and results cannot be remitted to supplementary information!!! Thus, the data and methods, and results sections must be incorporated and properly developed.

- The discussion section requires a profound review because i) it's a bit confusing; ii) mix results (which were not formally presented), with discussion and conclusions; iii) it is very extensive.

- The list of references is very extensive. There are a total of 128 references and 320 lines (6 pages of bibliography, in a total of 17 pages of text). A careful selection of the most representative works should be done. Some of the references can be considered as "grey literature", and for that reason avoided.

- Figures and tables. In general, the quality is quite good. However, some titles need to be improved (please see observations made in figures 1 to 7).

Author Response

REVIEWER 3

1- According to the authors, the main contribution of the paper is to explain the evolution of a sector of the rocky coast of Galicia, Spain (“Ria de Vigo”) since the last glacial period to nowadays, based on new data that are not well understood how they were obtained.

REPLY: Thank you for the comment of Reviewer 3. The new data are included in material and methods (Supplementary Material is deleted). The age calculated for the Cies Islands aeolianite led us to consider the previous premises and the revision of the data from aeolian sediments in the area (Ria de Vigo). We hope to resolve any doubts in this regard.

2- Although presenting an interesting subject the manuscript seems to be a poorly achieved synthesis of obtained data by the authors.

REPLY: Thank you for the comment of Reviewer 3. The manuscript is the result of a complex and laborious correlation of chronological, sedimentological, hypsometric and micropalaeontological data from numerous previous works which, as mentioned above, were only studied individually. Many of these data have been misinterpreted, confusing aeolian sediments with marine sediments, which forces a more extensive work (explaining where the most controversial aspects are and their resolution). Moreover, in this paper they are coherently contextualised and related to the sea level rise of the last transgression. All of this has led us to propose a novel hypothesis which highlights the presence of an important episode of coastal wind accretion, hitherto unidentified. This is why the manuscript is complex and sometimes dense. In this sense, we consider it to be a successful synthesis, as it questions previous premises and puts forward a new hypothesis that can be extrapolated to other areas at a local (Galician Rias), regional (Atlantic coast of SW Europe) and even global extent.

3- I do not think the manuscript is suitable for publication in its present form and needs major revision. Below I list some aspects that are on the basis of my decision. Authors will also be able to access the observations made up to page 8, on the submitted version. In general, the structure of the manuscript is rather confusing and consequently deserves a certain degree of reformulation. The title and abstract reflect the content of the article however the key words need tobe reviewed and be more objective.

REPLY: Thank you for the comment of Reviewer 3. The structure of the article has been reformulated for better understanding. The key words have been modified to be more objective about the subject matter.

4- The introduction should include the section 2. State of the question (why this name?) because it is here where the authors explores comprehensively the available literature (state of art) although the objectives in mind are not very clear.

REPLY: Thank you for the comment of Reviewer 3.At the suggestion of Reviewer 3, the introduction and the state of play have been reworded to improve the text.

5- Study area. This section needs to be improved regarding English form and grammar. Some geological and physical processes terms should also be reviewed.

REPLY: Thank you for the comment of Reviewer 3. At the suggestion of Reviewer 3, the Study Area section has been reworded in depth and simplified for better understanding.

6- The paper does not present in the body of the manuscript the sections related to methods and results. In a scientific paper the methodology and results cannot be remitted to supplementary information!!! Thus, the data and methods, and results sections must be incorporated and properly developed.

REPLY: Thank you for the comment of Reviewer 3. At the suggestion of Reviewer 3, sections (4) Material and Methods and (5) Results are included, with the deletion of Supplementary Material.

7- The discussion section requires a profound review because i) it's a bit confusing; ii) mix results (which were not formally presented), with discussion and conclusions; iii) it is very extensive.

REPLY: Thank you for the comment of Reviewer 3. The Discussion section is complex and dense. We have made the possible modifications for a better reading and understanding, trying not to transform the correlation of data into a meaningless mixture. The most relevant data on aeolian deposits and their chronology in the Ria de Vigo are presented before the Discussion and the reinforcement data included in the discussion are accompanied by a reference. The narrative of the text has been modified for improvement.  However, no sub-sections have been reduced as each of them is very important to understand each of the moments that affected the formation of the Ria de Vigo.

8- The list of references is very extensive. There are a total of 128 references and 320 lines (6 pages of bibliography, in a total of 17 pages of text). A careful selection of the most representative works should be done. Some of the references can be considered as "grey literature", and for that reason avoided.

REPLY: Thank you for the comment of Reviewer 3. At the suggestion of Reviewer 3, some bibliographical sources have been reduced, mainly those referring to works dealing with similar problems in different parts of the world.

9- Figures and tables. In general, the quality is quite good. However, some titles need to be improved (please see observations made in figures 1 to 7).

REPLY: Thank you for the comment of Reviewer 3. All these aspects have been modified. We sincerely thank Reviewer 3 as his constructive comments have been very helpful in understanding the reader's point of view. We hope that we have improved all the issues he has suggested.

Round 2

Reviewer 1 Report

Since all of my concerns have been well addressed, I am satisfied about the revision, which is much better than the previous one. Thus, I would like to recommend its acceptance. 

Author Response

Many thanks for the Reviewers' comments and suggestions, which, in our opinion, have led to a substantial improvement of the manuscript.

Reviewer 3 Report

The present version of the manuscript includes the suggestions made in the previous review. In this way, the paper can be accepted for publication.

Author Response

(The authors gave the same response as above.)
